# Exact manifold Gaussian Variational Bayes

## Abstract

We propose an optimization algorithm for Variational Inference (VI) in complex models. Our approach relies on natural gradient updates where the variational space is a Riemann manifold. We develop an efficient algorithm for Gaussian Variational Inference that implicitly satisfies the positive definite constraint on the variational covariance matrix. Our Exact manifold Gaussian Variational Bayes (EMGVB) provides exact but simple update rules and is straightforward to implement. Due to its black-box nature, EMGVB stands as a ready-to-use solution for VI in complex models. Over five datasets, we empirically validate our feasible approach on different statistical and econometric models, discussing its performance with respect to baseline methods.

## 1 Introduction

Although Bayesian principles are not new to Machine Learning (ML) (e.g. Mackay, 1992; 1995; Lampinen & Vehtari, 2001), it has been only recently that feasible methods boosted a growing use of Bayesian methods within the field (e.g. Zhang et al., 2018; Trusheim et al., 2018; Osawa et al., 2019; Khan et al., 2018b; Khan & Nielsen, 2018). In the typical ML settings the applicability of sampling methods for the challenging computation of the posterior is prohibitive, however approximate methods such as Variational Inference (VI) have been proved suitable and successful (Saul et al., 1996; Wainwright & Jordan, 2008; Hoffman et al., 2013; Blei et al., 2017). VI is generally performed with Stochastic Gradient Descent (SGD) methods (Robbins & Monro, 1951; Hoffman et al., 2013; Salimans & Knowles, 2014), boosted by the use of natural gradients (Hoffman et al., 2013; Wierstra et al., 2014; Khan et al., 2018b), and the updates often take a simple form (Khan & Nielsen, 2018; Osawa et al., 2019; Magris et al., 2022).

The majority of VI algorithms rely on the extensive use of models' gradients and the form of the variational posterior implies additional model-specific derivations that are not easy to adapt to a general, plug-and-play optimizer. Black box methods (Ranganath et al., 2014), are straightforward to implement and versatile use as they avoid model-specific derivations by relying on stochastic sampling (Salimans & Knowles, 2014; Paisley et al., 2012; Kingma & Welling, 2013). The increased variance in the gradient estimates as opposed to e.g. methods relying on the Reparametrization Trick (RT) (Blundell et al., 2015; Xu et al., 2019) can be alleviated with variance reduction techniques (e.g Magris et al., 2022).

Furthermore, the majority of existing algorithms do not directly address parameters' constraints. Under the typical Gaussian variational assumption, granting positive-definiteness of the covariance matrix is an acknowledged problem (e.g Tran et al., 2021a; Khan et al., 2018b; Lin et al., 2020). Only a few algorithms directly tackle the problem (Osawa et al., 2019; Lin et al., 2020), see Section 3. A recent approximate approach based on manifold optimization is provided by Tran et al. (2021a).

On the theoretical results of Khan & Lin (2017); Khan et al. (2018a) we develop an exact version of Tran et al. (2021a), resulting in an algorithm that explicitly tackles the positive-definiteness constraint for the variational covariance matrix and resembles the readily-applicable natural-gradient black-box framework of (Magris et al., 2022). For its implementation, we discuss recommendations and practicalities, show that EMGVB is of simple implementation, and demonstrate its feasibility in extensive experiments over four datasets, 12 models, and three competing VI optimizers.

In Section 2 we review the basis of VI, in Section 3 we review the Manifold Gaussian Variational Bayes approach and other related works, in Section 4 we discuss our proposed approach. Experi-

ments are found in Section 5, while Section 6 concludes. Appendices A, B complement the main discussion, Appendix C.4 reinforces and expands the experiments, Appendix D provides proofs.

## 2 VARIATIONAL INFERENCE

Variational Inference (VI) stands as a convenient and feasible approximate method for Bayesian inference. Let $y$ denote the data, $p(y|\theta)$ the likelihood of the data based on some model whose $k$-dimensional parameter is $\theta$. Let $p(\theta)$ be the prior distribution on $\theta$. In standard Bayesian inference the posterior is retrieved via the Bayes theorem as $p(\theta|y) = p(\theta)p(y|\theta)/p(y)$. As the marginal likelihood $p(y)$ is generally intractable, Bayesian inference is often difficult for complex models. Though the problem can be tackled with sampling methods, Monte Carlo techniques, although non-parametric and asymptotically exact may be slow, especially in high-dimensional applications (Salimans et al., 2015).

VI approximates the true unknown posterior with a probability density $q$ within a tractable class of distributions $\mathcal{Q}$, such as the exponential family. VI turns the Bayesian inference problem into that of finding the best variational distribution $q^\star \in \mathcal{Q}$ minimizing the Kullback-Leibler (KL) divergence from $q$ to $p(\theta|y)$: $q^\star = \arg\min_{q \in \mathcal{Q}} D_{\mathrm{KL}}(q||p(\theta|y))$. It can be shown that the KL minimization problem is equivalent to the maximization of the so-called Lower Bound (LB) on $\log p(y)$, (e.g. Tran et al., 2021b). In fixed-form variational Bayes, the parametric form of the variational posterior is set. The optimization problem accounts for finding the optimal variational parameter $\zeta$ parametrizing $q \equiv q_\zeta$ that maximizes the LB ($\mathcal{L}$), that is:

$$\zeta^\star = \arg\max_{\zeta \in \mathcal{Z}} \mathcal{L}(\zeta) := \int q_\zeta(\theta) \log \frac{p(\theta)p(y|\theta)}{q_\zeta(\theta)} d\theta = \mathbb{E}_{q_\zeta}\left[\log \frac{p(\theta)p(y|\theta)}{q_\zeta(\theta)}\right],$$

where $\mathbb{E}_q$ means that the expectation is taken with respect to the distribution $q_\zeta$, and $Z$ is the parameter space for $\zeta$.

The maximization of the LB is generally tackled with a gradient-descent method such as SGD (Robbins & Monro, 1951), ADAM Kingma & Ba (2014), or ADAGRAD Duchi et al. (2011). The learning of the parameter $\zeta$ based on standard gradient descent is however problematic as it ignores the information geometry of the distribution $q_\zeta$, is not scale invariant, unstable, and very susceptible to the initial values (Wierstra et al., 2014). SGD implicitly relies on the Euclidean norm for capturing the dissimilarity between two distributions, which can be a poor and misleading measure of dissimilarity (Khan & Nielsen, 2018). By using the KL divergence in place of the Euclidean norm, the SGD update results in the following natural gradient update:

$$\zeta_{t+1} = \zeta_t + \beta_t \left[\tilde{\nabla}_\zeta \mathcal{L}(\zeta)\right]\Big|_{\zeta=\zeta_t},$$

where $\beta_t$ is a possibly adaptive learning rate, and $t$ denotes the iteration. The above update results in improved steps towards the maximum of the LB when optimizing it for the variational parameter $\zeta$. The natural gradient $\tilde{\nabla}_\zeta \mathcal{L}(\zeta)$ is obtained by rescaling the euclidean gradient $\nabla_\zeta \mathcal{L}(\zeta)$ by the inverse of the Fisher Information Matrix (FIM) $\mathcal{I}_\zeta$, i.e. $\tilde{\nabla}_\zeta \mathcal{L}(\zeta) = \mathcal{I}_\zeta \nabla_\zeta \mathcal{L}(\zeta)$. For readability, we shall write $\mathcal{L}$ in place of $\mathcal{L}(\zeta)$.

A major issue in following this approach is that $\zeta$ is unconstrained. Think of a Gaussian variational posterior: under the above update, there is no guarantee that the covariance matrix is iteratively updated onto a symmetric and positive definite matrix. As discussed in the introduction, manifold optimization is an attractive possibility.

## 3 RELATED WORK

In Tran et al. (2021a), a $d$-dimensional Gaussian distribution $\mathcal{N}(\mu, \Sigma)$, provides the fixed-form of the variational posterior $q_\zeta = (\mu, \mathrm{vec}(\Sigma))$. There are no restrictions on $\mu$ yet the covariance matrix $\Sigma$ is constrained to the manifold $\mathcal{M}$ of symmetric positive-definite matric es, $\mathcal{M} = \left\{\Sigma \in \mathbb{R}^{d \times d} : \Sigma = \Sigma^\top, \Sigma \succ 0\right\}$, see e.g. (Abraham et al., 2012; Hu et al., 2020).

The exact form of the Fisher information matrix for the multivariate normal distribution is, e.g., provided in (Mardia & Marshall, 1984) and reads

$$\mathcal{I}_\zeta = \begin{pmatrix} \Sigma^{-1} & 0 \\ 0 & \mathcal{I}_\zeta(\Sigma) \end{pmatrix}, \qquad \mathcal{I}_\zeta(\Sigma)_{\sigma_{ij}, \sigma_{kl}} = \frac{1}{2}\text{tr}\left( \Sigma^{-1} \frac{\partial \Sigma}{\partial \sigma_{ij}} \Sigma^{-1} \frac{\partial \Sigma}{\partial \sigma_{kl}} \right). \qquad (1)$$

with $\mathcal{I}_\zeta(\Sigma)_{\sigma_{ij}, \sigma_{kl}}$ being the generic element of the $d^2 \times d^2$ matrix $\mathcal{I}_\zeta(\Sigma)$. The MGVB method relies on the approximation $\mathcal{I}_\zeta^{-1}(\Sigma) \approx (\Sigma^{-1} \otimes \Sigma^{-1})$, which leads to a convenient approximate form of the natural gradients of the lower bound with respect to $\mu$ and $\Sigma$, respectively computed as [1]

$$\tilde{\nabla}_\mu \mathcal{L}(\zeta) = \Sigma \nabla_\mu \mathcal{L}(\zeta) \qquad \text{and} \qquad \tilde{\nabla}_\Sigma \mathcal{L}(\zeta) \approx \text{vec}^{-1}\big((\Sigma \otimes \Sigma)\nabla_{\text{vec}(\Sigma)}\mathcal{L}(\zeta)\big) = \Sigma \nabla_\Sigma \mathcal{L}(\zeta)\Sigma, \quad (2)$$

where $\otimes$ denotes the Kronecker product. In virtue of the natural gradient definition, the first natural gradient is exact while the second is approximate. Thus, Tran et al. (2021a) adopt the following updates for the parameters of the variational posterior:

$$\mu = \mu + \beta \tilde{\nabla}_\mu \mathcal{L}(\zeta) \qquad \text{and} \qquad \Sigma = R_\Sigma(\beta \tilde{\nabla}_\Sigma \mathcal{L}(\zeta)), \qquad (3)$$

where $R_\Sigma(\cdot)$ denotes a suitable retraction for $\Sigma$ on the manifold $\mathcal{M}$. Momentum gradients can be used in place of plain natural ones. In particular, the momentum gradient for the update of $\Sigma$ relies on a vector transport granting that at each iteration the weighted gradient remains in the tangent space of the manifold $\mathcal{M}$. Refer to Section 4.2 for more information on retraction and vector transport. Besides the relationship between EMGVB and MGVB already discussed, a method of handling the positivity constraint in diagonal covariance matrices is VOGN optimizer (Khan et al., 2018b; Osawa et al., 2019). VOGN relates to the VON update (see Appendix B) as it indirectly updates $\mu$ and $\Sigma$ from the Gaussian natural parameters updates. Following a non-Black-Box approach, VOGN uses some theoretical results on the Gaussian distribution to recover an update for $\Sigma$ that involves the Hessian of the likelihood. Such Hessian is estimated as the samples' mean squared gradient, granting the non-negativity of the diagonal covariance update. Osawa et al. (2019) devise the computation of the approximate Hessian in a block-diagonal fashion within the layers of a Deep-learning model.

Lin et al. (2020) extend the above to handle the positive definiteness constraint by adding an additional term to the update rule for $\Sigma$, applicable to certain partitioned structures of the FIM. The retraction map in (Lin et al., 2020) is more general than (Tran et al., 2021a) and obtained through a different Riemann metric, from which MGVB is retrieved as a special case. As opposed to EMGVB, the use of the RT in (Lin et al., 2020) requires model-specific computation or auto-differentiation. See (Lin et al., 2021) for an extension on stochastic, non-convex problems. Lin et al. (2020) underline that in Tran et al. (2021a) the chosen form of the retraction is not well-justified as it is specific for the SPD matrix manifold, whereas the natural gradient is computed for the Gaussian manifold. An extensive discussion on this point and its relationship with the EMGVB optimizer here proposed is found in Appendix D.3.

Alternative methods that rely on unconstrained transformations (e.g. Cholesky factor) (e.g Tan, 2021), or on the adaptive adjustment of the learning rate (e.g. Khan & Lin, 2017) lie outside the manifold context here discussed. Among the methods that do not control for the positive definiteness constraint, the QBVI update (Magris et al., 2022) provides a comparable black-bock method that, despite other black-bock VI algorithms, uses exact natural gradients updates obtained without the computation of the FIM.

## 4 EXACT MANIFOLD GAUSSIAN VB

Consider a variational Gaussian distribution $q_\lambda$ with mean $\mu$ and positive-definite covariance matrix[2] $\Sigma$. Be $\lambda_1 = \Sigma^{-1}\mu$ and $\lambda_2 = -\frac{1}{2}\Sigma^{-1}$ its natural parameters and define $\lambda = (\lambda_1, \text{vec}(\lambda_2))$. The

---

[1]We present the MGVB optimizer by exactly following (Tran et al., 2021a). Lin et al. (2020) assert that in (Tran et al., 2021a) there is a typo as their $\mathcal{I}_\zeta^{-1}(\Sigma)$ term reads $\Sigma^{-1} \otimes \Sigma^{-1}$ in place of $2(\Sigma^{-1} \otimes \Sigma^{-1})$, which would lead to the actual natural gradient $2\Sigma\nabla_S\Sigma$ (e.g. Barfoot, 2020). While their observation is valid, we argue that the omission of the constant is embedded in the approximation, as it is also omitted from the implementation codes for MGVB, where $\tilde{\nabla}_\Sigma\mathcal{L}$ is computed as $\Sigma\nabla_S\Sigma$. To clarify, $\tilde{\nabla}_\Sigma\mathcal{L} = 2\Sigma\nabla_S\mathcal{L}\Sigma$ *is* an exact relationship, while $\tilde{\nabla}_\Sigma\mathcal{L} = \Sigma\nabla_S\mathcal{L}\Sigma$ not.

[2]This is the general case of practical relevance in applications, ruling out singular Gaussian distributions. For such peculiar distributions, $\Sigma$ is singular, $\Sigma^{-1}$ does not exist, and neither does the density. Though this might be theoretically interesting to develop, the discussion is here out of scope. Assuming $\Sigma$ to be positive-definite is not a restrictive and aligned with (Tran et al., 2021a)

corresponding mean or expectation parameters $m = (m_1, \text{vec}(m_2))$ are given by $m_1 = \mathbb{E}_{q_\lambda}[\theta] = \mu$ and $m_2 = \mathbb{E}_{q_\lambda}[\theta\theta^\top] = \mu\mu^\top + \Sigma$. When required, in place of the somewhat vague notation $\mathcal{L}$ whose precise meaning is to be inferred from the context, we shall use $\mathcal{L}(m)$ to explicitly denote the lower bound expressed in terms of the expectation parameter $m$, opposed to $\mathcal{L}(\lambda)$ expressed in terms of $\lambda$.

**Proposition 1** *For a differentiable function $\mathcal{L}$, and $q_\lambda$ being a Gaussian distribution with mean $\mu$ and covariance matrix S,*

$$\tilde{\nabla}_\mu \mathcal{L} = \Sigma \nabla_\mu \mathcal{L} \qquad \tilde{\nabla}_{\Sigma^{-1}} \mathcal{L} = -2\tilde{\nabla}_{\lambda_2} \mathcal{L} = -2\nabla_\Sigma \mathcal{L},$$

*where $\lambda_2 = -\frac{1}{2}\Sigma^{-1}$ denotes the second natural parameter of $q_\lambda$.*

The covariance matrix $\Sigma$ is positive definite, its inverse exists and it is as well symmetric and positive definite. Therefore $\Sigma^{-1}$ lies within the manifold $\mathcal{M}$ and can be updated with a suitable retraction algorithm as for $\Sigma$ in equation 3:

$$\Sigma^{-1} = R_{\Sigma^{-1}}\left(\beta\tilde{\nabla}_{\Sigma^{-1}}\mathcal{L}\right) = R_{\Sigma^{-1}}(-2\beta\nabla_\Sigma\mathcal{L}). \tag{4}$$

Opposed to the update in eq. 3, which relies on the approximation $\mathcal{I}_\zeta^{-1}(\Sigma) \approx \Sigma^{-1} \otimes \Sigma^{-1}$, for tacking a positive-definite update of $\Sigma$, we target at updating $\Sigma^{-1}$, for which its natural gradient is available in an exact form, by primarily exploiting the duality between the gradients in the natural and expectation parameter space (Appendix D.1, eq. 25) that circumvents the computation of the FIM.

For coherency with the literature on VI for Bayesian deep learning (e.g. Ranganath et al., 2014, among many others), we specify the variational posterior in terms of the covariance matrix $\Sigma$, but update $\Sigma^{-1}$. Yet nothing prevents specifying $q_\lambda$ in terms of its precision matrix $\Sigma^{-1}$, as is often the case in Bayesian statistics textbooks, in which this case, the update 4 corresponds to an update for the actual variational precision parameter.

For updating $\mu$ is reasonable to adopt plain SGD-like step driven by the natural parameter $\tilde{\nabla}_\mu\mathcal{L} = \Sigma\nabla_\mu\mathcal{L}$, as in (Tran et al., 2021a). We refer to the following update rules as Exact Manifold Gaussian Variational Bayes, or shortly EMGVB,

$$\mu_{t+1} = \mu_t + \beta\Sigma\nabla_\mu\mathcal{L}_t \qquad \text{and} \qquad \Sigma_{t+1}^{-1} = R_{\Sigma_t^{-1}}(-2\beta\nabla_\Sigma\mathcal{L}_t), \tag{5}$$

where the gradients w.r.t. $\mathcal{L}$ are intended as evaluated at the current value of the parameters, e.g. $\nabla_\Sigma\mathcal{L}_t = \nabla_\Sigma\mathcal{L}|_{\mu=\mu_t, \Sigma=\Sigma_t}$. With respect to the MGVB update of Tran et al. (2021a), there are no approximations, e.g. regarding the FIM, yet the cost of updating $\Sigma^{-1}$ appears to be that of introducing an additional inversion for retrieving $\Sigma$ that is involved in the EMGVB update for $\mu$. In the following Section, we show that with a certain gradient estimator such an inversion is irrelevant. Furthermore, in Appendix B we point out that a covariance matrix inversion is implicit in both MGVB and EMGVB due to the second-order form of the retraction and also show that the update for $\mu$ is optimal in the sense therein specified.

## 4.1 Implementation

We elaborate on how to evaluate the gradients $\nabla_\Sigma\mathcal{L}$ and $\nabla_\mu\mathcal{L}$. We follow the Black-box approach (Ranganath et al., 2014) under which such gradients are approximated via Monte Carlo (MC) sampling and rely on function queries only. The implementation of the EMGVB updates does not require the model's gradient to be specified nor to be computed numerically, e.g. with backpropagation. By use of the so-called log-derivative trick (see e.g. (Ranganath et al., 2014)) it is possible to evaluate the gradients of the LB as an expectation with respect to the variational distribution. In particular, for a generic differentiation variable $\zeta$,

$$\nabla_\zeta\mathcal{L}(\zeta) = \mathbb{E}_{q_\lambda}[\nabla_\zeta[\log q_\zeta(\theta)]\, h_\zeta(\theta)], \quad \text{where} \quad h_\zeta(\theta) = \log\left[\frac{p(\theta)p(y|\theta)}{q_\zeta(\theta)}\right].$$

In the EMGVB context with $q \sim \mathcal{N}(\mu, \Sigma)$, $\zeta = (\mu, \text{vec}(\Sigma))$ and $\mathcal{L}(\zeta) = \mathcal{L}(\mu, \Sigma)$. The gradient of the $\mathcal{L}$ w.r.t. $\zeta$ evaluated at $\zeta = \zeta_t$ can be easily estimated using $S$ samples from the variational posterior through the unbiased estimator

$$\nabla_\zeta\mathcal{L}(\zeta_t) = \nabla_\zeta\mathcal{L}(\zeta)|_{\zeta=\zeta_t} \approx \frac{1}{S}\sum_{s=1}^{S}[\nabla_\zeta[\log q_\zeta(\theta_s)]\, h_\zeta(\theta_s)]|_{\zeta=\zeta_t}, \quad \theta_s \sim \mathcal{N}(\mu_t, S_t) \tag{6}$$

where the $h$-function is evaluated in the current values of the parameters, i.e. in $\zeta_t = (\mu_t, \text{vec}(\Sigma_t))$. For a Gaussian distribution $q \sim \mathcal{N}(\mu, \Sigma)$ it can be shown that (e.g. Wierstra et al., 2014; Magris et al., 2022):

$$\nabla_\mu \log q(\theta) = \Sigma^{-1}(\theta - \mu), \tag{7}$$

$$\nabla_\Sigma \log q(\theta) = -\frac{1}{2}\Big(\Sigma^{-1} - \Sigma^{-1}(\theta - \mu)(\theta - \mu)^\top \Sigma^{-1}\Big), \tag{8}$$

Equations 7, 8 along with 6 and Proposition 1 immediately lead to the feasible natural gradients estimators:

$$\tilde{\nabla}_\mu \mathcal{L}(\zeta_t) \approx \Sigma_t \hat{\nabla}_\mu \mathcal{L}(\zeta_t) = \frac{1}{S} \sum_{s=1}^{S} [(\theta_s - \mu_t) h_{\zeta_t}(\theta_s)], \tag{9}$$

$$\tilde{\nabla}_{\Sigma^{-1}} \mathcal{L}(\zeta_t) \approx -2\hat{\nabla}_\Sigma \mathcal{L}(\zeta_t) = \frac{1}{S} \sum_{s=1}^{S} \Big[\Big(\Sigma_t^{-1} - \Sigma_t^{-1}(\theta_s - \mu_t)(\theta_s - \mu_t)^\top \Sigma_t^{-1}\Big) h_{\zeta_t}(\theta_s)\Big]. \tag{10}$$

As for the MGVB update, the EMGVB update applies exclusively to Gaussian variational posteriors, yet no constraints are imposed on the parametric form of $p$. When considering a Gaussian prior, the implementation of the EMGVB update can take advantage of some analytical results leading to MC estimators of reduced variance, namely implemented over the log-likelihood $\log p(y|\theta_s)$ rather than the $h$-function.

In Appendix D.2, we show that, under a Gaussian prior specification, the above updates can be also implemented in terms of the model likelihood than in terms of the $h$-function. The general form of the EMGVB updates then writes:

$$\tilde{\nabla}_\mu \mathcal{L}(\zeta_t) \approx c_{\mu_t} + \frac{1}{S} \sum_{s=1}^{S} [(\theta_s - \mu_t) \log f(\theta_s)] \tag{11}$$

$$\tilde{\nabla}_{\Sigma^{-1}} \mathcal{L}(\zeta_t) \approx C_{\Sigma_t} + \frac{1}{S} \sum_{s=1}^{S} \Big[\Big(\Sigma_t^{-1} - \Sigma_t^{-1}(\theta_s - \mu_t)(\theta_s - \mu_t)^\top \Sigma_t^{-1}\Big) \log f(\theta_s)\Big] \tag{12}$$

where, (i) if $p$ is Gaussian $C_{\Sigma_t} = -\Sigma_t^{-1} + \Sigma_0^{-1}$, $c_{\mu_t} = -\Sigma_t \Sigma_0^{-1}(\mu_t - \mu_0)$ and $\log f(\theta_s) = \log p(y|\theta_s)$, (ii) if $p$ is Gaussian or not $C_{\Sigma_t} = c_{\mu_t} = 0$ and $\log f(\theta_s) = h_{\zeta_t}(\theta_s)$.

and $\theta_s \sim q_{\zeta_t} = \mathcal{N}(\mu_t, \Sigma_t)$, $s = 1, \ldots, S$. $\log(y|\theta_s)$ and $h_{\zeta_t}(\theta_s)$ respectively denote the model likelihood and the $h$-function evaluated in $\theta_s$. Note that the latter depends on $t$ as it involves the variational posterior, evaluated at the parameters at the value of the parameters for iteration $t$. $p$ denotes the prior. It is clear that under the Gaussian specification for $p$ the MC estimator is of reduced variance, compared to the general one based on the $h$-function. Note that the log-likelihood case does not involve an additional inversion for retrieving $\Sigma$ in $c_\mu$, as $\Sigma$ is anyway required in the second-order retraction (for both MGVB and EMGVB). This aspect is further discussed Appendix B. For Inverting $\Sigma^{-1}$ we suggest inverting the Cholesky factor $L^{-1}$ of $\Sigma^{-1}$ and compute $\Sigma$ as $LL^\top$. This takes advantage of the triangular form of $L^{-1}$ which can be inverted with back-substitution, which is $k^3/3$ flops cheaper than inverting $\Sigma^{-1}$, but still $\mathcal{O}(k^3)$. $L$ is furthermore used for generating the draws $\theta_s$ as either $\theta_s = \mu + L\varepsilon$ or $\theta_s = \mu + L^{-\top}\varepsilon$ with $\varepsilon$, with $\varepsilon \sim \mathcal{N}(0, \boldsymbol{I})$. As outlined in Appendix A, we devise the use of control variates to reduce the variance of the stochastic gradient estimators.

Though the lower bound is not directly involved in EMGVB updates, it can be naively estimated at each iteration as

$$\hat{\mathcal{L}}_t = \frac{1}{S} \sum_{s=1}^{S} [p(\theta) + \log p(y|\theta) - \log q_\zeta(\theta)]. \tag{13}$$

As discussed in Appendix A, $\hat{\mathcal{L}}_t$ is needed for terminating the optimization routine, verifying anomalies in the algorithm works (the LB should actually increase and converge) and comparing EMGVB with MGVB, see Section 5.

## 4.2 RETRACTION AND VECTOR TRANSPORT

Aligned with Tran et al. (2021a), we adopt the retraction method advanced in (Jeuris et al., 2012) for the manifold $\mathcal{M}$ of symmetric and positive definite matrices

$$R_{\Sigma^{-1}}(\xi) = \Sigma^{-1} + \xi + \frac{1}{2}\xi\Sigma\xi, \quad \text{where } \xi \in T_{\Sigma^{-1}}\mathcal{M}, \tag{14}$$

with $\xi$ being the rescaled natural gradient $\beta\tilde{\nabla}_{\Sigma^{-1}}\mathcal{L} = -2\beta\nabla_{\Sigma}\mathcal{L}$. In practice, whenever applicable, as e.g. in the retraction, for granting the symmetric from of a matrix (or gradient matrix) $S$, we compute $S$ as $1/2(S + S^{\top})$. Vector transport is as well easily implemented by

$$\mathcal{T}_{\Sigma_t^{-1}\to\Sigma_{t+1}^{-1}}(\xi) = E\xi E^{\top}, \quad \text{where } E = \left(\Sigma_{t+1}^{-1}\Sigma_t\right)^{\frac{1}{2}}, \xi \in T_{\Sigma^{-1}}\mathcal{M}. \tag{15}$$

We refer to the Manopt toolbox (Boumal et al., 2014) for the practical details of implementing the above two algorithms in a numerically stable fashion. This translates into the momentum gradients

$$\tilde{\nabla}_{\Sigma^{-1}}^{\text{mom.}}\mathcal{L}_{t+1} = \omega\,\mathcal{T}_{\Sigma_t^{-1}\to\Sigma_{t+1}^{-1}}\left(\tilde{\nabla}_{\Sigma^{-1}}^{\text{mom.}}\mathcal{L}_t\right) + (1-\omega)\tilde{\nabla}_{\Sigma^{-1}}\mathcal{L}_{t+1}, \tag{16}$$

$$\tilde{\nabla}_{\mu}^{\text{mom.}}\mathcal{L}_{t+1} = \omega\,\tilde{\nabla}_{\mu}^{\text{mom.}}\mathcal{L}_t + (1-\omega)\tilde{\nabla}_{\mu}\mathcal{L}_t, \tag{17}$$

where the weight $0 < \omega < 1$ is a hyper-parameter.

The attentive reader may recognize the adoption of the form of retraction and parallel transport obtained from the SPD (matrix) manifold on the natural gradient obtained from the Gaussian manifold. This apparent inconsistency in mixing elements of different manifold structures is discussed in Appendix D.3. We show that, from a learning perspective, the discrepancy between the form of the SPD manifold Riemann gradient $\Sigma^{-1}\nabla_{\Sigma^{-1}}\Sigma^{-1} = -\nabla_{\Sigma}$ and the natural gradient $\tilde{\nabla}_{\Sigma^{-1}}\mathcal{L} = -2\nabla_{\Sigma}\mathcal{L} = -2\Sigma^{-1}\nabla_{\Sigma^{-1}}\Sigma^{-1}$ is absorbed in the learning rate $\beta$. In particular, our update rule can be derived within a fully consistent SPD manifold setting by updating $\left(\mu, 2\Sigma^{-1}\right)$.

In the above view, we can now further clarify that the wording "Exact" in EMGVB is twofold. (i) In Tran et al. (2021a) the natural gradient $\Sigma\nabla_{\Sigma}\Sigma$ is in place of the actual one $2\Sigma\Sigma^{-1}\Sigma$, whose corresponding one for $\Sigma^{-1}$ is the one that EMGVB actually adopts. (ii) Even by the adoption of the actual natural gradient $-2\Sigma^{-1}\nabla_{\Sigma^{-1}}\mathcal{L}\Sigma$, the use of the SPD retraction and vector transport forms 14,15, as of Tran et al. (2021a), are not well-justified: in Appendix D.3 these are justified, and EMGVB is shown to be a consistent approach. Note that EMGVB is exact in the sense of the above, yet still approximate in absolute terms due to the use of retraction. Retractions are approximate forms of the exponential map tracing back vectors on the tangent space to the manifold, which is generally cumbersome transform to compute and impractical (e.g. Absil et al., 2009; Hu et al., 2020).

Algorithm 1 summarizes the EMGVB update for the Gaussian prior-variational posterior case. Computational aspects are discussed in Appendix B.2.

---

**Algorithm 1** EMGVB implementation

---

1: Set hyper-parameters: $0 < \beta, \omega < 1$, $S$
2: Set the type of gradient estimator, i.e. function $\log f(\theta_s)$
3: Set prior $p(\theta)$, likelihood form $p(y|\theta)$, and initial values $\mu, \Sigma^{-1}$
4: $t = 1$, Stop $=$ `false`
5: Generate: $\theta_s \sim q_{\mu_1, \Sigma_1}, s = 1 \ldots S$
6: Compute: $\hat{g}_{\mu} = \Sigma\hat{\nabla}_{\mu}\mathcal{L}, \hat{g}_{\Sigma^{-1}} = -2\hat{\nabla}_{\Sigma}\mathcal{L}$       ▷ eqs. 11,12
7: $m_{\mu} = \hat{g}_{\mu}, m_{\Sigma^{-1}} = \hat{g}_{\Sigma^{-1}}$       ▷ initialize momentum
8: **while** Stop $=$ `true` **do**
9:   $\mu = \mu + \beta m_{\mu}$       ▷ EMGVB update for $\mu$
10:   $\Sigma_{\text{old}}^{-1} = \Sigma^{-1}, \quad \Sigma^{-1} = R_{\Sigma^{-1}}(\beta m_{\Sigma^{-1}})$       ▷ EMGVB update for $\Sigma^{-1}$
11:   Generate: $\theta_s \sim q_{\mu_t, \Sigma_t}, s = 1 \ldots S$
12:   Compute: $\hat{g}_{\mu}, \hat{g}_{\Sigma^{-1}}$       ▷ as in line 6
13:   $m_{\mu} = \omega m_{\mu} + (1-\omega)\hat{g}_{\mu}$       ▷ eq. 16
14:   $m_{\Sigma^{-1}} = \mathcal{T}_{\Sigma_{\text{old}}^{-1}\to\Sigma^{-1}}(m_{\Sigma^{-1}}) + (1-\omega)\hat{g}_{\Sigma^{-1}}$       ▷ eq. 17
15:   Compute: $\hat{\mathcal{L}}_t$       ▷ eq. 13
16:   $t = t + 1$, Stop $= f_{\text{exit}}\left(\bar{\mathcal{L}}, P, t_{\max}\right)$       ▷ see Appendix A
17: **end while**

---

## 4.3 FURTHER CONSIDERATIONS

Along with the choice of the gradient estimator and the use of momentum, there are other aspects of relevance in the implementation of EMGVB. Details are discussed in Appendix A.

## 4.4 ISOTROPIC PRIOR

For mid-sized to large-scale problems, the prior is commonly specified as an isotropic Gaussian of mean $\mu_0$, often $\mu_0 = \mathbf{0}$, and covariance matrix $\Sigma_0^{-1} = \tau \mathbf{I}$, with $\tau > 0$ a scalar precision parameter. The covariance matrix of the variational posterior can be either diagonal or not. Whether a full co-variance specification ($d^2 - d$ parameters) can provide additional degrees of freedom that can gauge models' predictive ability, a diagonal posterior ($d$ parameters) can be practically and computationally convenient to adopt e.g. in large-sized problems. The diagonal-posterior assumption is largely adopted in Bayesian inference and VI (e.g. Blundell et al., 2015; Ganguly & Earp, 2021; Tran et al., 2021b) and Bayesian ML applications (e.g. Kingma & Welling, 2013; Graves, 2011; Khan et al., 2018b; Osawa et al., 2019), in Appendix A we provide a block-diagonal variant.

### 4.4.1 ISOTROPIC PRIOR AND DIAGONAL GAUSSIAN POSTERIOR

Assume a $d$-variate diagonal Gaussian variational specification, that is $q \sim \mathcal{N}(\mu, \Sigma)$ with $\mathrm{diag}(\Sigma) = \boldsymbol{\sigma}^2$, $\Sigma_{ij} = 0$, for $i, j = 1, \ldots, d$ and $i \neq j$. In this case, $\Sigma^{-1} = \mathrm{diag}(1/\boldsymbol{\sigma}^2)$, where the division is intended element-wise, and $\nabla_\Sigma \mathcal{L} = \mathrm{diag}(\nabla_{\boldsymbol{\sigma}^2} \mathcal{L})$, is now a $d \times 1$ vector. Updating $\Sigma^{-1}$ amounts to updating $\boldsymbol{\sigma}^{-2}$: the natural gradient retraction-based update for $\boldsymbol{\sigma}^{-2}$ is now based on the equality $\tilde{\nabla}_{\boldsymbol{\sigma}^{-2}} \mathcal{L} = -2\nabla_{\boldsymbol{\sigma}^2} \mathcal{L}$, so that the general-case EMGVB update reads

$$\boldsymbol{\sigma}_{t+1}^{-2} = R_{\boldsymbol{\sigma}_{t+1}^{-2}}\left(-2\beta \nabla_{\boldsymbol{\sigma}^2} \mathcal{L}\right) \qquad \text{and} \qquad \mu_{t+1} = \mu_t + \boldsymbol{\sigma}_t^2 \odot \beta \nabla_\mu \mathcal{L} \qquad (18)$$

where $\odot$ denotes the element-wise product. The corresponding MC estimators for the gradients are

$$-2\hat{\nabla}_{\boldsymbol{\sigma}^2} \mathcal{L} \approx c_{\boldsymbol{\sigma}_t^2} + \boldsymbol{\sigma}^{-2} \odot \frac{1}{S} \sum_{s=1}^{S} \left[\left(\mathbf{1}_d - (\theta_s - \mu_t)^2 \odot \boldsymbol{\sigma}^{-2}\right) \log p_t(y|\theta_s)\right] \qquad (19)$$

$$\boldsymbol{\sigma}^2 \odot \hat{\nabla}_\mu \mathcal{L} \approx c_{\mu_t} + \frac{1}{S} \sum_{s=1}^{S} [(\theta_s - \mu_t) \log p_t(y|\theta_s)], \qquad (20)$$

where $c_{\boldsymbol{\sigma}_t^2} = -\boldsymbol{\sigma}^{-2} + \tau$, $c_{\mu_t} = \tau \boldsymbol{\sigma}^2 \odot (\mu - \mu_0)$, $\theta_s \sim \mathcal{N}(\mu, \mathrm{diag}(\boldsymbol{\sigma}^2))$, $s = 1 \ldots, S$, $(\theta_s - \mu_t)^2$ is intended element-wise, and $\mathbf{1}_d = (1, \ldots, 1)^\top \in \mathbb{R}^d$. In the Gaussian case with a general diagonal covariance matrix, retrieving $\boldsymbol{\sigma}^2$ from the updated $\boldsymbol{\sigma}^{-2}$ is inexpensive as $\sigma_i^2 = 1/\sigma_i^{-2}$, indicating that in this context the use of the $h$-function estimator is never advisable.

### 4.4.2 ISOTROPIC PRIOR AND FULL GAUSSIAN POSTERIOR

Because of the full form of the covariance matrix, this case is rather analogous to the general one. In particular, factors $c_{\mu_t}$ and $c_{\Sigma_t}$ in eq. 29 are replaced by (i) $c_{\Sigma_t} = -\Sigma^{-1} + \tau$, $c_{\mu_t} = \tau \Sigma(\mu_t - \mu_0)$ or (ii) $c_{\Sigma_t} = 0$, $c_{\mu_t} = 0$, respectively under the Gaussian-prior case ($\log f_t(\theta_s) = \log p(y|\theta_s)$) and the general one ($\log f_t(\theta_s) = h(\theta)$). The MC estimators 7 and 8 apply: (i) leads to an estimator of reduced variance, while (ii) is identical to the general case.

## 5 EXPERIMENTS

We validate and explore the empirical validity and feasibility of our suggested optimizer over four datasets and 12 models. These include logistic regression (Labor dataset), different volatility models on S&P 500 returns (Volatility dataset), and linear regression on Stock indexes (Istanbul data). Details on the datasets and models are summarized in Appendix 11. The main baseline for model comparison is the MGVB optimizer and (sequential) MCMC estimates representative of the true posterior. Additionally, we also include results related to the QBVI optimizer Magris et al. (2022). In this section, we report synthetic results on two tasks: logistic regression (classification) and volatility

modeling with the FIGARCH model (regression). Results on the other datasets and models appear in Appendix C.4. Matlab codes are available at github.com/blinded.

The bottom rows in Figures 1, 5 clearly show that our results align with the sampling-based MCMC results and with the Maximum Likelihood (ML) estimates. Whereas marginal posterior approximations are rather close between EMGVB and MGVB, the top row in Figures 1, 5 show that the parameter learning is qualitatively different. The panels in Figure 1 depict the LB optimization process across the iterations. In a diagonal posterior setting, MGVB is exact and aligns with EMGVB (middle panel), however for non-diagonal posteriors, EMGVB's lower bound shows an improved convergence rate on both the training and test sets (left and right panels respectively). Furthermore, we observe that the adoption of the $h$-function estimator has a minimal impact. From the point of view of standard performance measures, Figure 2 shows that compared to MGVB, at early iterations, EMGVB displays a steeper growth in model accuracy, precision, recall, and f1-score both on the training test and test set. Ultimately EMGVB and MGVB measures converge to the same value, yet the exact nature of the EMGVB update leads to convergence after approximately 200 iterations on the training set as opposed to 500 for MGVB. A similar behavior is observed for the FIGARCH(1,1,1) model, the top row of Figure 5.

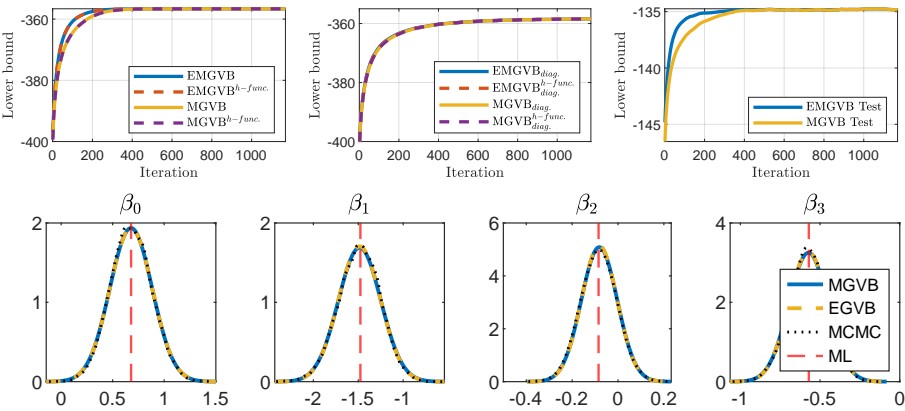

Figure 1: Top row: lower bound optimization. Bottom row: variational posteriors (for four of the eight parameters).

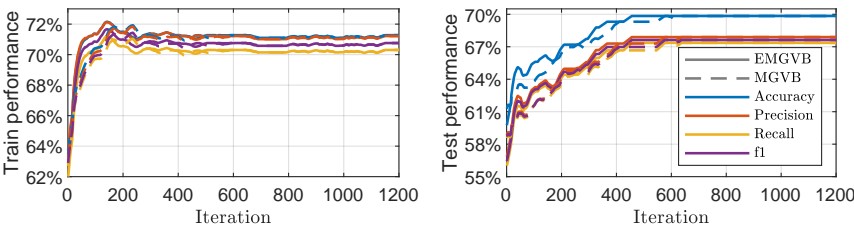

Figure 2: EMGVB and MGVB performance on the Labour dataset across the iterations.

Tables 1 and 2 report such performance measures for the optimized variational posterior along with the value of the maximized lower bound. EMGVB is very close to the baselines and well-aligned with the MCMC and ML estimates, which, along with the estimates (in Table 3 for the logistic regression), show that EMGVB converges towards the same LB maximum, with a comparable predictive ability with respect to the alternatives. It is thus not surprising that the estimates, performance metrics, and value of the optimized LB are similar across the optimizers: they all converge to the *same* minimum but in a *qualitatively different* way. Also estimated variational covariance matrix for the full-covariance cases, closely replicates the one from the MCMC chain (see tables 4, 6). For the diagonal cases, MCMC and ML covariance matrices are not suitable for a direct comparison (see Appendix C.3). The sanity check in Figure 4 furthermore shows that the learning of either the

mean and covariance variation parameters is smooth and steady without wigglings or anomalies. As expected, the non-diagonal version leads to faster convergence while the use of the $h$-function estimator slightly stabilizes the learning process. In Table 7 we also show that the impact the number of MC samples $S$ has on posterior means, likelihood, performance measures, and the optimized lower bound is minor for both the training and test phases.

| | Train | | | | | Test | | | | |
|---|---|---|---|---|---|---|---|---|---|---|
| | $\mathcal{L}(\theta^\star)$ | Accuracy | Precision | Recall | f1 | $\mathcal{L}(\theta^\star)$ | Accuracy | Precision | Recall | f1 |
| EMGVB | -356.642 | 0.713 | 0.712 | 0.703 | 0.708 | -134.814 | 0.698 | 0.679 | 0.674 | 0.676 |
| MGVB | -356.642 | 0.713 | 0.712 | 0.703 | 0.708 | -134.814 | 0.698 | 0.679 | 0.674 | 0.676 |
| QBVI | -356.642 | 0.713 | 0.712 | 0.703 | 0.708 | -134.804 | 0.698 | 0.679 | 0.674 | 0.676 |
| MCMC | | 0.711 | 0.710 | 0.701 | 0.706 | | 0.698 | 0.679 | 0.674 | 0.676 |
| ML | | 0.709 | 0.708 | 0.699 | 0.704 | | 0.709 | 0.708 | 0.699 | 0.704 |

Table 1: Optimizers' performance for the Labor data on the train and test sets. See Appendix C.4 for extended results, including the use of the $h-$function estimator, diagonal and block-diagonal covariance specifications.

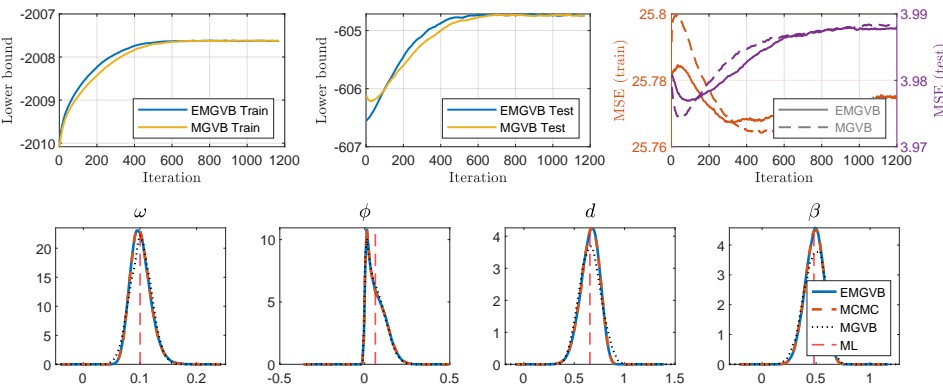

Figure 3: FIGARCH(1,1,1) model. Top row: lower bound optimization. Bottom row: variational marginals.

| | | | | | Train | | Test | |
|---|---|---|---|---|---|---|---|---|
| | $\bar\omega$ | $\phi$ | $d$ | $\beta$ | $\mathcal{L}(\theta^\star)$ | MSE | $\mathcal{L}(\theta^\star)$ | MSE |
| EMGVB | 0.100 | 0.059 | 0.663 | 0.481 | -2007.62 | 25.773 | -604.74 | 3.988 |
| MGVB | 0.100 | 0.059 | 0.663 | 0.481 | -2007.62 | 25.773 | -604.72 | 3.988 |
| QBVI | 0.100 | 0.059 | 0.663 | 0.480 | -2007.62 | 25.771 | -604.73 | 3.988 |
| MCMC | 0.100 | 0.062 | 0.656 | 0.480 | | 25.784 | | 3.979 |
| ML | 0.099 | 0.060 | 0.669 | 0.483 | | 25.767 | | 3.996 |

Table 2: Optimizers' estimates and performance for the FIGARCH(1,1,1) model on the Volatility dataset.

## 6   CONCLUSION

Within a Gaussian variational framework, we propose an algorithm based on manifold optimization to guarantee the positive-definite constraint on the covariance matrix, employing exact analytical solutions for the natural gradients. Our black-box optimizer results in a ready-to-use solution for VI, scalable to structured covariance matrices, that can take advantage of control variates, momentum, and alternative forms of the stochastic gradient estimator. We show the feasibility of our solution on a multitude of models. Our approach aligns with sampling methods and provides advantages over state-of-the-art baselines. Future research may investigate the applicability of our approach to a broader set of variational distributions, explore the advantages and limitations of the black-box framework, or attempt at addressing the online inversion bottleneck of manifold-based VI.

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

# A FURTHER CONSIDERATIONS ON EMGVB IMPLEMENTATION

## A.1 VARIANCE REDUCTION

As EMGVB does not involve model gradients the use of the reparametrization trick (RT) (Blundell et al., 2015) is not immediate. While eq. 5 would generally hold, the form of the EMGVB gradient estimators under the RT would differ from eqs. 7, 8: we develop EMGVB as a general and ready-to-use solution for VI that does not require model-specific derivations, yet one may certainly enable the RT within EMGVB. Though the use of the RT is quite popular in VI and ML as it empirically yields more accurate estimates of the gradient of the variational objective than alternative approaches (Mohamed et al., 2020), note that the variance of the RT estimator can be higher than that of the score-function estimator and the path-wise RT estimator is not necessarily preferable (Xu et al., 2019; Mohamed et al., 2020). Not less importantly, note that the use of the score estimator is broader as it does not require $\log p(y|\theta)$ to be differentiable.

Control Variates (CV) stand as a simple and effective approach for reducing the variance of the MC gradient estimator, e.g. (Paisley et al., 2012). The CV estimator

$$\frac{1}{S} \sum_{s=1}^{S} \nabla_\zeta [\log q(\theta_s)](\log p(y|\theta_s) - c),$$

is unbiased for the expected gradient, but of equal or smaller variance that the naive MC one. For $i = 1, 2$ the optimal $c_i$ minimizing the variances of the CV estimator is

$$c^\star = \mathrm{Cov}(\nabla_\zeta[\log q(\theta)] \log p(y|\theta), \nabla_\zeta \log q(\theta)) / \mathrm{Var}(\nabla_\zeta \log q(\theta)). \tag{21}$$

By enabling CVs, $S$ can be tuned to balance the estimates' variance and computational performance. In Table 7 we asses that for logistic regression values of $S$ as little as 10 appear satisfactory, yet if the iterative computation of the log-likelihood is not prohibitive we suggest the adoption of a more generous value, e.g. $S \approx 100$. Magris et al. (2022) furthermore shows that the denominator in 21 is analytically tractable for a Gaussian $q$, reducing the variance of estimated $c^\star$ and thus improving the overall CVs' efficiency. If model gradients are available one may use CV along with the RT to further enhance the efficiency of the expected gradient estimation.

## A.2 LB SMOOTHING AND STOPPING CRITERION

The stochastic nature of the gradient estimator introduces some noise in the estimated LB $\hat{\mathcal{L}}$ that can violate its expected non-decreasing behavior across the iterations. By setting window of size $w$ we rather consider the moving average on the LB, $\bar{\mathcal{L}}_t = 1/w \sum_{i=1}^{t} \hat{\mathcal{L}}_{t-i+1}$, whose variance is reduced and behavior stabilized. By keeping track of $\max \bar{\mathcal{L}}$ we terminate the the learning after $\max \bar{\mathcal{L}}$ did not improved for $P$ iterations (patience parameter) or after a maximum number of iteration ($t_{\max}$) is reached (stopping criterion $f_{\mathrm{exit}}$ function in Algorithm 1).

## A.3 CONSTRAINTS ON MODEL PARAMETERS

EMGVB assumes a Gaussian variational posterior, that is the parameters are unbounded and defined over the entire real line. Assuming that a model parameter $\theta$ is required to lie on a support $\mathcal{S}$, to impose such a constraint it suffices to identify a feasible transform $T : \mathbb{R} \to \mathcal{S}$ and apply the EMGVB update to the unconstrained parameter $\psi = T^{-1}(\theta)$. Certainly, by applying VI on $\psi$ we require that the variational posterior assumption holds for $\psi$ rather than $\theta$. The actual distribution for $\theta$ under a Gaussian variational $\psi$ can be computed (or approximated with a sampling method) as $\mathcal{N}\big(T^{-1}(\theta); \mu, \Sigma\big)|\det(J_{T^{-1}}(\theta))|$, with $J_{T^{-1}}$ the Jacobin of the inverse transform (Kucukelbir et al., 2015). *Example.* For the GARCH(1,1) model (see Section 5) the intercept $\omega$, the autoregressive coefficient of the lag-one squared return $\alpha$ and moving-average coefficient $\beta$ of the lag-one conditional variances need to satisfy the stationarity conditions $\alpha + \beta < 1$ and $\omega > 0, \alpha \geq 0, \beta \geq 0$. Such conditions are unfeasible under a Gaussian variational approximation: we estimate the unconstrained parameters $\psi_\omega, \psi_\alpha, \psi_\beta$, where $\omega = T(\psi_\omega), \alpha = T(\psi_\alpha)(1 - T(\psi_\beta)), \beta = T(\psi_\alpha)T(\psi_\beta)$ with $T(x) = \exp(x)/(1 + \exp(x))$ for $x$ real, on which Gaussian's prior-posterior assumptions apply.

## A.4 Gradient clipping

Especially for low values of $S$, and even more if a variance control method is not adopted, the stochastic gradient estimate may be poor and the offset from its actual value large. This may result in updates whose magnitude is too big either in a positive or negative direction. Especially at early iterations, and with poor initial values, this issue may e.g. cause complex roots in eq. 15. At each iteration $t$, to control for the magnitude of the stochastic gradient $\hat{g}_t$ we rescale its $\ell_2$-norm $||\hat{g}_t||$ whenever it is larger than a fixed threshold $l_{\max}$ by replacing $\hat{g}_t$ with $\hat{g}_t l_{\max}||\hat{g}_t||$, which preserves its norm. Gradient clipping can be either applied to the gradients $\hat{\nabla}_\mu$, $\hat{\nabla}_\Sigma$ or to the natural gradients $\Sigma\hat{\nabla}_\mu$, $-2\hat{\nabla}_\Sigma$ and in any case before obtaining momentum gradients. We suggest applying gradient clipping readily to $\hat{\nabla}_\mu$, $\hat{\nabla}_\Sigma$ to promptly mitigate the impact that far-from-the-mean estimates may have on successive computations.

## A.5 Adaptive learning rate

It is convenient to adopt an adaptive learning rate or scheduler for decreasing $\beta$ after a certain number of iterations. Typical options are that of reducing $\beta$ by a certain factor (e.g. 0.2) every set number of iterations (e.g. 100), or decrease it after iteration $t'$ e.g. by setting $\beta_t = \min -(\beta, \beta\frac{t'}{t})$, where $t'$ is a fraction (e.g. 0.7) of the maximum number of iterations $t_{\max}$ allowed before the LB optimization is stopped.

## A.6 Classification vs. regression

We point out that the EMGVB framework is applicable to both regression and classification problems. In generic DL classification problems, predictions are based on the class of maximum probability which is computed by applying a softmax function at the last layer returning to the probability $p_i(c_j)$ of a certain class $c_j$ for the $i$-th sample, $i = 1, \ldots, M$. From these probabilities it is straightforward to compute the model log-likelihood as $\sum_{i=1}^{M} y_{i,c_{\text{true}}} \log p_i(c_{\text{true}})$, with $y_{i,c_{\text{true}}}$ representing the one-key-hot encoding of the $i$-th sample, whose true class is $c_{\text{true}}$. For regression, the parametric form of $\log p(y|\theta)$ is clearly different and model-specific (e.g. regression with normal errors as opposed to Poisson regression, with the latter being feasible as the use of the score estimator does not require the likelihood to be differentiable). Note that however additional parameters may enter into play besides the ones involved in the back-bone forward model: e.g. for regression with normal errors tackled with an artificial neural network, the Gaussian-form likelihood involves the regression variance, which is an additional parameter over the network's ones, or for Student-t errors the degree of freedom parameter $\nu$ (with the constraint $\nu > 2$). See the application in Appendix C.3.

## A.7 Mean-field variant

Assume that for a $d$-variate model the Gaussian variational posterior is factorized as

$$q_\varsigma(\theta) = q_{\varsigma_1}(\theta_1)q_{\varsigma_2}(\theta_2), \ldots, q_{\varsigma_h}(\theta_h) = \prod_{i=1}^{h} q_{\varsigma_h}(\theta_h),$$

with $h \leq k$. If $h = d$ this corresponds to a full-diagonal case where each $\theta_i$ is a scalar and the covariance between $\varsigma_1, \ldots, \varsigma_k$ is ignored. If $h < d$, the variational covariance matrix $\Sigma$ of $q_\varsigma$ corresponds to a block-diagonal matrix, and some of the $\theta$s are indeed vectors. In any case, the expected gradients with respect to each block of parameters can be computed independently, given the scalars $h_{\varsigma_h}(\theta)$ or $\log p(y|\theta)$, depending on whether the $h$-function estimator is used. For a Gaussian prior, its covariance matrix can be diagonal, full, block-diagonal with a structure matching or not that of $S$. Eqs. 11, 12, with the condition 29 can be used as a starting point to derive case-specific EMGVB variants based on the form of the prior covariance.

Algorithm 2 summarizes the case with an isotropic Gaussian prior of zero-mean and variance $\tau$, using the gradient estimator based on the log-likelihood: $\mu_i$, $\Sigma_i$ ($\Sigma_i^{-1}$) respectively denote the mean and covariance (precision) matrix of the $i$-th block of $\Sigma$. In this case, the block-wise natural gradients

are estimated as

$$\Sigma\hat{\nabla}_{\mu_i}\mathcal{L} = -S\tau^{-1}\mu_i + \frac{1}{S}\sum_{s=1}^{S}[(\theta_{s_i} - \mu_i)\log p(y|\theta_s)],$$

$$-2\hat{\nabla}_{\Sigma_i}\mathcal{L} = -\Sigma_i + \text{diag}(\tau^{-1}) + \frac{1}{S}\sum_{s=1}^{S}\Big[\Big(\Sigma_i^{-1} - \Sigma_i^{-1}(\theta_{s_i} - \mu_i)(\theta_{s_i} - \mu_i)^\top\Sigma_i^{-1}\Big)\log p(y|\theta_s)\Big],$$

where $\theta_s$ is a sample from the variational posterior. $\theta_s$ can be obtained by concatenating marginal samples from each block, $\theta_s = [\theta_{s_1}, \ldots, \theta_{s_h}]$, with $\theta_{s_i} \sim q_{\mu_i, \Sigma_i}, i = 1, \ldots, h$

---

**Algorithm 2** EMGVB for a block-diagonal covariance matrix (prior with zero-mean and covariance matrix $\tau\boldsymbol{I}$)

---

1: Set hyper-parameters: $0 < \beta, \omega < 1, S$
2: Set the type of gradient estimator, i.e. function $\log f(\theta_s)$
3: Set prior $p(\theta; 0, \tau)$, likelihood $p(y|\theta)$, and initial values $\mu, \Sigma^{-1}$
4: $t = 1$, Stop $=$ `false`
5: Generate: $\theta_s = [\theta_{s_1}, \ldots, \theta_{s_h}], \quad \theta_{s_i} \sim q_{\mu_i, \Sigma_i}, s = 1 \ldots S, i = 1, \ldots, h$
6: Compute: $\log p(y|\theta_s)$
7: **for** $i = 1, \ldots, h$ **do**
8:     Compute: $\hat{g}_{\mu_i} = \Sigma_i\hat{\nabla}_{\mu_i}\mathcal{L}, \quad \hat{g}_{\Sigma_i^{-1}} = -2\hat{\nabla}_{\Sigma_i}\mathcal{L}$
9:     $m_{\mu_i} = \hat{g}_{\mu_i}, \quad m_{\Sigma_i^{-1}} = \hat{g}_{\Sigma_i^{-1}}$
10: **end for**
11: **while** Stop $=$ `true` **do**
12:     $\hat{\mathcal{L}} = 0$
13:     **for** $i = 1, \ldots, h$ **do**
14:         $\mu_i = \mu_i + \beta m_{\mu_i}$
15:         $\Sigma_{\text{old},i}^{-1} = \Sigma_i^{-1}, \quad \Sigma_i^{-1} = R_{\Sigma_i^{-1}}\Big(\beta m_{\Sigma_i^{-1}}\Big)$
16:     **end for**
17:     Generate: $\theta_s = [\theta_{s_1}, \ldots, \theta_{s_h}], \quad \theta_{s_i} \sim q_{\mu_i, \Sigma_i}, s = 1 \ldots S, i = 1, \ldots, h$
18:     Compute: $\log p(y|\theta_s), \quad \log p(\theta_s)$
19:     **for** $i = 1, \ldots, h$ **do**
20:         Compute: $\hat{g}_{\mu_i}, \hat{g}_{\Sigma_i^{-1}}$
21:         $m_{\mu_i} = \omega m_{\mu_i} + (1 - \omega)\hat{g}_{\mu_i}$
22:         $m_{\Sigma_i^{-1}} = \mathcal{T}_{\Sigma_{\text{old},i}^{-1} \to \Sigma_i^{-1}}\Big(m_{\Sigma_i^{-1}}\Big) + (1 - \omega)\hat{g}_{\Sigma_i^{-1}}$
23:         $\hat{\mathcal{L}}_t = \mathcal{L} + \frac{1}{S}\log p(\theta_s) + \frac{1}{S}\log p(y|\theta_s) - \frac{1}{S}\log q_{m_i, \Sigma_i}(\theta_{s_i})$
24:         $t = t + 1$, Stop $= f_{\text{exit}}(\bar{\mathcal{L}}, P, t_{\max})$
25:     **end for**
26: **end while**

---

# B   OPTIMALITY AND EFFICIENCY

## B.1   OPTIMALITY

Several authors (e.g. Khan & Lin, 2017; Khan et al., 2018a) obtained update rules for VI by developing over SGD-like updates for the natural parameters of the variational posterior. By updating the natural parameter $\lambda$ and exploiting its definition, it is relatively simple to recover the update rules for $\mu$ and $\Sigma$. Indeed from the SGD update for the natural parameter $\lambda_{t+1} = \lambda_t + \beta\tilde{\nabla}_\lambda\mathcal{L}(\lambda)$ it follows

$$\mu_{t+1} = \Sigma_{t+1}\Big[\Sigma_t^{-1}\mu_t + \beta\Big[\tilde{\nabla}_\lambda\mathcal{L}(\lambda_t)\Big]\Big]$$
$$= \Sigma_{t+1}\Big[\big(\Sigma_t^{-1} - 2\beta[\nabla_\Sigma\mathcal{L}(\lambda_t)]\big)\mu_t + \beta[\nabla_\mu\mathcal{L}(\lambda_t)]\Big], \tag{22}$$

and

$$\Sigma_{t+1}^{-1} = \Sigma_t^{-1} - 2\beta[\nabla_\Sigma\mathcal{L}(\lambda_t)] \tag{23}$$

By replacing $\Sigma_t^{-1} - 2\beta[\nabla_\Sigma\mathcal{L}(\lambda_t)]$ with $\Sigma_{t+1}^{-1}$ in the update for $\mu$, Khan & Lin (2017); Khan et al. (2018a) obtain

$$\mu_{t+1} = \mu_t + \beta\Sigma_{t+1}[\nabla_\mu\mathcal{L}(\lambda_t)]. \tag{24}$$

Eq. 23 does not apply to the EMGVB update as the update for $\Sigma^{-1}$ is carried out with retraction and does not result from an SGD update of the natural parameter $\lambda_2 = -\frac{1}{2}\Sigma_{t+1}^{-1}$, yet the form of equation 22 does apply. We refer to eq. 23 as an indirect update since derived from the natural parameter update. Note that, the $\mu$ update exploits $\Sigma_{t+1}^{-1}$, resulting in a one-step forward-looking rule. It is relevant to investigate whether the update 22 is preferable to the EMGVB update for $\mu$. Intuitively one might expect that eq. 22 is preferable as it somewhat readily exploits the updated $\Sigma_{t+1}$ value as soon as it becomes available. The following theorem however proves that this is not the case, a proof is provided in Appendix D.4.

**Theorem 1** *For the Gaussian distribution with parameters $\zeta = (\mu, vec(\Sigma))$ the optimization problem*

$$\zeta_{t+1} = \arg\min_{\zeta} \langle \zeta, \nabla_{\zeta}\mathcal{L}(\zeta_t) \rangle + \frac{1}{\beta}D_{\mathrm{KL}}(p_\zeta || p_{\zeta_t}),$$

*where $\langle \cdot, \cdot \rangle$ denotes the inner product and $\nabla_{\zeta}\mathcal{L}(\zeta_t) = (\nabla_{\mu}\mathcal{L}(\zeta), vec(\nabla_{\Sigma}\mathcal{L}(\zeta)))|_{\zeta=\zeta_t}$, is convex with respect to $\zeta$. The optimum update for $\mu$ is available in closed form and analogous to that of the EMGBV update.*

The objective in Theorem 1, is that of the mirror descent developed by Nemirovskij & Yudin (1983), where a non-Euclidean geometry is induced by considering a penalized optimization obtained through proximity function such as the Bergman divergence, which equals the KL divergence for exponential-family distributions. In this regard see (Raskutti & Mukherjee, 2015).

Following Theorem 1, the update for $\mu$ in eq. 5 is optimal in terms of the above objective and perhaps counter-intuitively the indirect forward-looking update 22 is proved to provide non-optimal steps toward the maximization of the lower bound. The EMGVB update for $\mu$ is thus preferable over the alternative of recovering an indirect update rule for $\mu$ starting from an SGD update on the natural parameter as e.g. in (Khan & Lin, 2017; Khan et al., 2018a): in the above terms of Theorem 1, the EMGVB update for $\mu$ is the best one could take.

### B.2 COMPUTATIONAL ASPECTS

In terms of computational complexity, the exact EMGVB implementation is at no additional cost. Actually, the cost of computing the natural gradient in EMGVB as $-2\nabla_{\Sigma}^{-1}\mathcal{L}$ is cheaper than the one in MGVB, $\Sigma\nabla_{\Sigma}\mathcal{L}\Sigma$, $O(k^3)$ operations for each matrix multiplication. However, both MGVB and EMGVB share a cumbersome matrix inversion.

Going back to eqs. 12 and 11, it is noticeable that under the most general estimator based on the $h$-function, $\Sigma$ is not involved in any computation, neither in the gradient involved in the updated for $\Sigma^{-1}$ nor in that for $\mu$, suggesting that implicitly the EMGVB optimization routine does not require the inversion of $\Sigma^{-1}$. The above point however ignores that the update for $\Sigma^{-1}$ is masked by the underlying retraction. For the retraction form in eq. 14, both $\Sigma^{-1}$ and its inverse $\Sigma$ are needed, thus implying a matrix inversion at every iteration. That is, the covariance matrix inversion is implicit in MGVB and EMGVB methods, which both require $\Sigma^{-1}$ and $\Sigma$ at every iteration (with little surprise, as the form of the retraction is a second-order approximation of the exponential map). With the $h$-function estimator even though neither eq. 7 nor 8 involve $\Sigma$, the inversion of $\Sigma^{-1}$ is still necessary, as $\Sigma$ is required in retraction. Similarly, the adoption of the log-likelihood estimator under the Gaussian regime in eq. 29, is not computationally more expensive than the $h$-function case, as $\Sigma$, involved in $c_{\mu_t}$, is anyway required in retraction. As outlined in Section 4, $\Sigma$ can be conveniently recovered from the Cholesky factor of $\Sigma^{-1}$, with a lesser number of flips. Lastly, if $\Sigma^{-1}$ ($\Sigma$) is diagonal, the inversion is trivial and, when applicable, eq. 11 is preferred.

## C EXPERIMENTS

### C.1 RESULTS FOR THE LABOUR DATA

|  | $\beta_0$ | $\beta_1$ | $\beta_2$ | $\beta_3$ | $\beta_4$ | $\beta_5$ | $\beta_6$ | $\beta_7$ |
|---|---|---|---|---|---|---|---|---|
| MGVB | 0.678 | $-1.485$ | $-0.082$ | $-0.573$ | 0.494 | $-0.637$ | 0.608 | 0.041 |
| EMGVB | 0.679 | $-1.485$ | $-0.082$ | $-0.573$ | 0.494 | $-0.637$ | 0.608 | 0.041 |
| QBVI | 0.679 | $-1.485$ | $-0.082$ | $-0.573$ | 0.494 | $-0.637$ | 0.608 | 0.041 |
| MGVB$^\dagger$ | 0.678 | $-1.487$ | $-0.084$ | $-0.574$ | 0.493 | $-0.638$ | 0.609 | 0.048 |
| EMGVB$^\dagger$ | 0.678 | $-1.487$ | $-0.084$ | $-0.574$ | 0.493 | $-0.638$ | 0.609 | 0.048 |
| QBVI$^\dagger$ | 0.678 | $-1.487$ | $-0.084$ | $-0.574$ | 0.493 | $-0.638$ | 0.609 | 0.048 |
| MGVB$^{\text{diag.}}$ | 0.576 | $-1.430$ | $-0.056$ | $-0.541$ | 0.493 | $-0.638$ | 0.593 | 0.105 |
| EMGVB$^{\text{diag.}}$ | 0.578 | $-1.431$ | $-0.057$ | $-0.542$ | 0.493 | $-0.637$ | 0.593 | 0.104 |
| QBVI$^{\text{diag.}}$ | 0.579 | $-1.432$ | $-0.057$ | $-0.542$ | 0.493 | $-0.637$ | 0.593 | 0.103 |
| MGVB$^{\dagger\,\text{diag.}}$ | 0.573 | $-1.430$ | $-0.057$ | $-0.542$ | 0.493 | $-0.635$ | 0.593 | 0.109 |
| EMGVB$^{\dagger\,\text{diag.}}$ | 0.575 | $-1.431$ | $-0.057$ | $-0.543$ | 0.493 | $-0.635$ | 0.593 | 0.107 |
| QBVI$^{\dagger\,\text{diag.}}$ | 0.576 | $-1.431$ | $-0.057$ | $-0.543$ | 0.493 | $-0.635$ | 0.593 | 0.107 |
| MCMC | 0.675 | $-1.484$ | $-0.084$ | $-0.570$ | 0.493 | $-0.639$ | 0.608 | 0.051 |
| ML | 0.679 | $-1.476$ | $-0.085$ | $-0.571$ | 0.485 | $-0.625$ | 0.599 | 0.045 |

Table 3: Parameters' estimates for the labour dataset. Top: posterior means, middle: variances, bottom: covariances ($\times 10^3$). $^\dagger$ denotes the use of the $h$-function gradient estimator, $^{\text{diag}}$ the use of a diagonal variational posterior.

MGVB

|  |  | $\beta_0$ | $\beta_1$ | $\beta_2$ | $\beta_3$ | $\beta_4$ | $\beta_5$ | $\beta_6$ | $\beta_7$ |
|---|---|---|---|---|---|---|---|---|---|
| | $\beta_0$ | | $-1.639$ | $-0.848$ | $-0.697$ | 0.163 | 0.262 | 0.192 | $-2.709$ |
| | $\beta_1$ | $-1.634$ | | 0.298 | 1.425 | $-0.451$ | $-0.020$ | 0.084 | $-0.118$ |
| | $\beta_2$ | $-0.810$ | 0.288 | | 0.413 | 0.066 | 0.050 | $-0.064$ | $-0.109$ |
| EMGVB | $\beta_3$ | $-0.700$ | 1.393 | 0.406 | | 0.039 | 0.128 | $-0.189$ | $-0.397$ |
| | $\beta_4$ | 0.077 | $-0.361$ | 0.127 | 0.083 | | $-0.135$ | $-0.259$ | $-0.191$ |
| | $\beta_5$ | 0.312 | $-0.051$ | 0.008 | 0.068 | $-0.144$ | | $-1.322$ | $-0.611$ |
| | $\beta_6$ | 0.176 | 0.073 | $-0.021$ | $-0.096$ | $-0.230$ | $-1.202$ | | $-0.028$ |
| | $\beta_7$ | $-2.809$ | $-0.048$ | $-0.103$ | $-0.381$ | $-0.218$ | $-0.633$ | $-0.073$ | |

ML

|  |  | $\beta_0$ | $\beta_1$ | $\beta_2$ | $\beta_3$ | $\beta_4$ | $\beta_5$ | $\beta_6$ | $\beta_7$ |
|---|---|---|---|---|---|---|---|---|---|
| | $\beta_0$ | | $-1.578$ | $-0.800$ | $-0.684$ | 0.067 | 0.318 | 0.194 | $-2.701$ |
| | $\beta_1$ | $-1.510$ | | 0.274 | 1.378 | $-0.422$ | $-0.027$ | 0.074 | $-0.084$ |
| | $\beta_2$ | $-0.791$ | 0.249 | | 0.393 | 0.102 | 0.024 | $-0.058$ | $-0.093$ |
| MCMC | $\beta_3$ | $-0.654$ | 1.307 | 0.381 | | 0.067 | 0.117 | $-0.164$ | $-0.352$ |
| | $\beta_4$ | 0.047 | $-0.414$ | 0.111 | 0.072 | | $-0.172$ | $-0.265$ | $-0.109$ |
| | $\beta_5$ | 0.334 | $-0.049$ | 0.006 | 0.089 | $-0.177$ | | $-1.286$ | $-0.613$ |
| | $\beta_6$ | 0.172 | 0.111 | $-0.036$ | $-0.137$ | $-0.266$ | $-1.282$ | | $-0.097$ |
| | $\beta_7$ | $-2.603$ | $-0.085$ | $-0.097$ | $-0.349$ | $-0.119$ | $-0.590$ | $-0.102$ | |

Table 4: Parameters' covariance matrices for the labour dataset. Entries are multiplied by $10^2$.

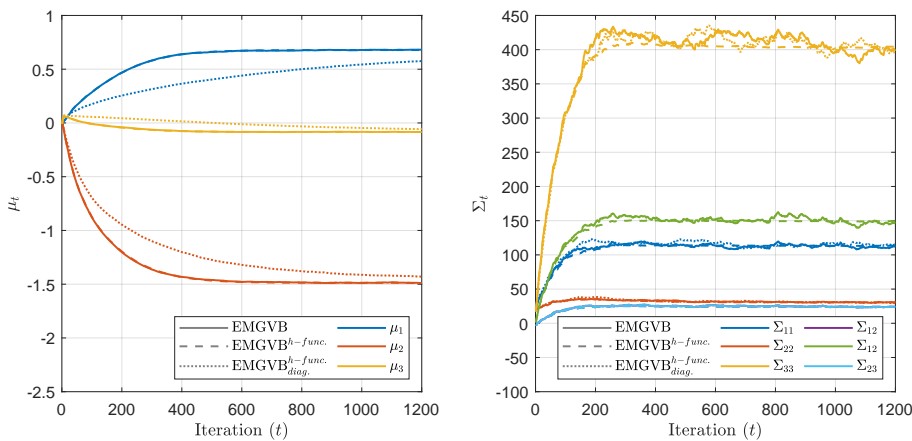

Figure 4: Parameter learning across the iterations under different variants of the EMGVB algorithm for the labour dataset.

|  | Train | | | | | |
|---|---|---|---|---|---|---|
|  | $\mathcal{L}(\theta^\star)$ | $\log p(y|\theta^\star)$ | Accuracy | Precision | Recall | f1 |
| EMGVB | $-356.642$ | $-332.992$ | 0.713 | 0.712 | 0.703 | 0.708 |
| MGVB | $-356.642$ | $-332.991$ | 0.713 | 0.712 | 0.703 | 0.708 |
| QBVI | $-356.642$ | $-332.992$ | 0.713 | 0.712 | 0.703 | 0.708 |
| EMGVB$^\dagger$ | $-356.635$ | $-332.990$ | 0.711 | 0.710 | 0.701 | 0.706 |
| MGVB$^\dagger$ | $-356.635$ | $-332.990$ | 0.711 | 0.710 | 0.701 | 0.706 |
| QBVI$^\dagger$ | $-356.635$ | $-332.990$ | 0.711 | 0.710 | 0.701 | 0.706 |
| EMGVB$^{\text{diag.}}$ | $-358.423$ | $-333.124$ | 0.711 | 0.710 | 0.701 | 0.706 |
| MGVB$^{\text{diag.}}$ | $-358.428$ | $-333.129$ | 0.711 | 0.710 | 0.701 | 0.706 |
| QBVI$^{\text{diag.}}$ | $-358.419$ | $-333.121$ | 0.711 | 0.710 | 0.701 | 0.706 |
| EMGVB$^{\dagger\,\text{diag.}}$ | $-358.421$ | $-333.127$ | 0.709 | 0.708 | 0.700 | 0.704 |
| MGVB$^{\dagger\,\text{diag.}}$ | $-358.427$ | $-333.133$ | 0.709 | 0.708 | 0.700 | 0.704 |
| QBVI$^{\dagger\,\text{diag.}}$ | $-358.418$ | $-333.124$ | 0.709 | 0.708 | 0.700 | 0.704 |
| MCMC |  | $-332.990$ | 0.711 | 0.710 | 0.701 | 0.706 |
| ML |  | $-332.983$ | 0.709 | 0.708 | 0.699 | 0.704 |

|  | Test | | | | | |
|---|---|---|---|---|---|---|
|  | $\mathcal{L}(\theta^\star)$ | $\log p(y|\theta^\star)$ | Accuracy | Precision | Recall | f1 |
| EMGVB | $-134.814$ | $-113.854$ | 0.698 | 0.679 | 0.674 | 0.676 |
| MGVB | $-134.814$ | $-113.854$ | 0.698 | 0.679 | 0.674 | 0.676 |
| QBVI | $-134.804$ | $-113.854$ | 0.698 | 0.679 | 0.674 | 0.676 |
| EMGVB$^\dagger$ | $-134.805$ | $-113.855$ | 0.698 | 0.679 | 0.674 | 0.676 |
| MGVB$^\dagger$ | $-134.805$ | $-113.855$ | 0.698 | 0.679 | 0.674 | 0.676 |
| QBVI$^\dagger$ | $-134.805$ | $-113.855$ | 0.698 | 0.679 | 0.674 | 0.676 |
| EMGVB$^{\text{diag.}}$ | $-136.864$ | $-114.171$ | 0.667 | 0.644 | 0.640 | 0.642 |
| MGVB$^{\text{diag.}}$ | $-136.871$ | $-114.178$ | 0.667 | 0.644 | 0.640 | 0.642 |
| QBVI$^{\text{diag.}}$ | $-136.857$ | $-114.167$ | 0.667 | 0.644 | 0.640 | 0.642 |
| EMGVB$^{\dagger\,\text{diag.}}$ | $-136.876$ | $-114.220$ | 0.667 | 0.644 | 0.640 | 0.642 |
| MGVB$^{\dagger\,\text{diag.}}$ | $-136.883$ | $-114.228$ | 0.667 | 0.644 | 0.640 | 0.642 |
| QBVI$^{\dagger\,\text{diag.}}$ | $-136.871$ | $-114.217$ | 0.667 | 0.644 | 0.640 | 0.642 |
| MCMC |  | $-113.838$ | 0.698 | 0.679 | 0.674 | 0.676 |
| ML |  | $-113.800$ | 0.709 | 0.708 | 0.699 | 0.704 |

Table 5: Models' performance for the labour data on the train and test sets. $^\dagger$ denotes the use of the $h$-function gradient estimator, $^{\text{diag}}$ the use of a diagonal variational posterior.

|  | $\beta_0$ | $\beta_1$ | $\beta_2$ | $\beta_3$ | $\beta_4$ | $\beta_5$ | $\beta_6$ | $\beta_7$ |
|---|---|---|---|---|---|---|---|---|
| EMGVB | 4.242 | 5.465 | 0.601 | 1.479 | 1.220 | 1.931 | 1.879 | 4.477 |
| MGVB | 4.240 | 5.574 | 0.615 | 1.506 | 1.288 | 2.127 | 1.927 | 4.393 |
| QBVI | 4.255 | 5.484 | 0.603 | 1.484 | 1.224 | 1.938 | 1.886 | 4.493 |
| EMGVB$^\dagger$ | 4.068 | 5.415 | 0.594 | 1.464 | 1.290 | 2.043 | 1.968 | 4.385 |
| MGVB$^\dagger$ | 4.069 | 5.416 | 0.598 | 1.469 | 1.287 | 2.058 | 1.974 | 4.387 |
| QBVI$^\dagger$ | 4.068 | 5.415 | 0.594 | 1.464 | 1.290 | 2.043 | 1.968 | 4.385 |
| EMGVB$^{\text{diag.}}$ | 0.909 | 3.356 | 0.255 | 0.895 | 0.988 | 0.955 | 0.994 | 1.286 |
| MGVB$^{\text{diag.}}$ | 0.909 | 3.356 | 0.255 | 0.894 | 0.988 | 0.955 | 0.993 | 1.286 |
| QBVI$^{\text{diag.}}$ | 0.910 | 3.363 | 0.255 | 0.896 | 0.989 | 0.956 | 0.995 | 1.288 |
| EMGVB$^{\dagger\,\text{diag.}}$ | 0.871 | 3.391 | 0.252 | 0.897 | 1.051 | 0.999 | 1.006 | 1.313 |
| MGVB$^{\dagger\,\text{diag.}}$ | 0.871 | 3.390 | 0.252 | 0.897 | 1.051 | 1.000 | 1.006 | 1.313 |
| QBVI$^{\dagger\,\text{diag.}}$ | 0.872 | 3.394 | 0.252 | 0.897 | 1.052 | 1.000 | 1.007 | 1.315 |
| MCMC | 3.967 | 5.364 | 0.589 | 1.406 | 1.274 | 2.071 | 1.966 | 4.248 |
| ML | 4.079 | 5.436 | 0.589 | 1.457 | 1.276 | 2.059 | 1.960 | 4.393 |

Table 6: Variances of the parameters for the labour dataset. Entries are multiplied by $10^2$. $^\dagger$ denotes the use of the $h$-function gradient estimator, $^{\text{diag}}$ the use of a diagonal variational posterior. For a discussion on the diagonal case see Appendix C.3.

| $S$ | $t$ | $\beta_0$ | $\beta_1$ | $\beta_2$ | $\beta_3$ | $\beta_4$ | $\beta_5$ | $\beta_6$ | $\beta_7$ | $\mathcal{L}(\theta_0)$ |
|---|---|---|---|---|---|---|---|---|---|---|
| 10 | 5.103 | 0.687 | −1.498 | −0.086 | −0.576 | 0.500 | −0.636 | 0.605 | 0.037 | −430.216 |
| 20 | 6.984 | 0.682 | −1.498 | −0.085 | −0.571 | 0.499 | −0.641 | 0.607 | 0.043 | −429.626 |
| 30 | 8.137 | 0.684 | −1.493 | −0.087 | −0.573 | 0.495 | −0.640 | 0.609 | 0.043 | −434.863 |
| 50 | 9.015 | 0.680 | −1.492 | −0.085 | −0.576 | 0.494 | −0.643 | 0.611 | 0.047 | −435.890 |
| 75 | 11.788 | 0.682 | −1.489 | −0.085 | −0.575 | 0.492 | −0.642 | 0.611 | 0.045 | −436.282 |
| 100 | 14.652 | 0.682 | −1.490 | −0.085 | −0.574 | 0.493 | −0.639 | 0.609 | 0.044 | −436.924 |
| 150 | 22.862 | 0.680 | −1.488 | −0.085 | −0.574 | 0.492 | −0.639 | 0.609 | 0.046 | −436.759 |
| 200 | 24.169 | 0.681 | −1.487 | −0.085 | −0.574 | 0.492 | −0.638 | 0.608 | 0.046 | −437.036 |
| 300 | 31.708 | 0.680 | −1.488 | −0.085 | −0.575 | 0.493 | −0.638 | 0.609 | 0.046 | −436.529 |

| | | Train | | | | |
|---|---|---|---|---|---|---|
| $S$ | $\mathcal{L}(\theta^\star)$ | $\log p(y\|\theta^\star)$ | Accuracy | Precision | Recall | f1 |
| 10 | −356.687 | −332.991 | 0.711 | 0.710 | 0.701 | 0.706 |
| 20 | −356.667 | −332.994 | 0.713 | 0.712 | 0.703 | 0.708 |
| 30 | −356.653 | −332.990 | 0.709 | 0.708 | 0.699 | 0.704 |
| 50 | −356.645 | −332.988 | 0.713 | 0.712 | 0.703 | 0.708 |
| 75 | −356.642 | −332.992 | 0.713 | 0.712 | 0.703 | 0.708 |
| 100 | −356.640 | −332.988 | 0.711 | 0.710 | 0.701 | 0.706 |
| 150 | −356.640 | −332.990 | 0.711 | 0.710 | 0.701 | 0.706 |
| 200 | −356.638 | −332.990 | 0.713 | 0.712 | 0.703 | 0.708 |
| 300 | −356.639 | −332.989 | 0.711 | 0.710 | 0.701 | 0.706 |

| | | Test | | | | |
|---|---|---|---|---|---|---|
| $S$ | $\mathcal{L}(\theta^\star)$ | $\log p(y\|\theta^\star)$ | Accuracy | Precision | Recall | f1 |
| 10 | −134.865 | −113.910 | 0.698 | 0.679 | 0.674 | 0.676 |
| 20 | −134.899 | −113.856 | 0.698 | 0.679 | 0.674 | 0.676 |
| 30 | −134.793 | −113.827 | 0.698 | 0.679 | 0.674 | 0.676 |
| 50 | −134.805 | −113.838 | 0.698 | 0.679 | 0.674 | 0.676 |
| 75 | −134.814 | −113.854 | 0.698 | 0.679 | 0.674 | 0.676 |
| 100 | −134.810 | −113.876 | 0.698 | 0.679 | 0.674 | 0.676 |
| 150 | −134.791 | −113.845 | 0.698 | 0.679 | 0.674 | 0.676 |
| 200 | −134.788 | −113.840 | 0.698 | 0.679 | 0.674 | 0.676 |
| 300 | −134.832 | −113.842 | 0.698 | 0.679 | 0.674 | 0.676 |

Table 7: Estimated parameters and performance measures on the labour dataset for EMGVB (full-posterior) for different sizes of the number of MC draws for the estimation of the stochastic gradients $S$. $t$ refers to the run-time per iteration (in milliseconds), $\mathcal{L}(\theta_0)$ to the LB evaluated at the initial parameters. For each $S$, a common random seed used.

Our second set of experiments involves the estimation of several GARCH-family volatility models. The models in Table 8 differ for the number of estimated parameters, the form of the likelihood function (which can be quite complex as for the FIGARCH models), and constraints imposed on the parameters. Besides the GARCH-type models, we include the well-known linear HAR model for realized volatility (Corsi, 2009). We performed a preliminary study for retaining only relevant models, e.g. we observed that for a GARCH(1,0,2) $\beta_2$ is not significant, so we trained a GARCH(1,0,1), or that the autoregressive coefficient of the squared innovations is always significant only at lag one, so we did not consider further lags for $\alpha$. For $\alpha, \beta, \gamma$ we restricted the search up to lag 2. Except for HAR's parameter $\beta_3$, all the parameters of all the models are statistically significant under standard ML at $5\%$. Note that the aim of this experiment is that of applying VI and EMGVB to the above class of models, not to discuss their empirical performance or forecasting ability. For the reader unfamiliar with the above (standard) models, discussion and notation we refer e.g. to the accessible introduction of Teräsvirta (2009).

As for the Labour data, we report the values of the smoothed lower bound computed at the optimized parameter $\mathcal{L}(\hat{\theta}^\star)$, the model's log-likelihood in the estimates posterior parameter $p(y|\theta^\star)$ and the MSE between the fitted values and squared daily returns, used as a volatility proxy. Details on the data and hyperparameters are provided in Table 11.

Figures 6 provide sample illustrations of the lower bound maximization for the GJR(1,1,1) model (perhaps the most used and effective in applications beyond the standard GARCH(1,0,1)) and the FIGARCH(1,1,1) model, the most complex one among our selection due to the form of the likelihood, constraints, and econometric interpretation. In general, beyond figure 6, we witness a slight but consistent improved convergence of the lower bound towards its maximum the train set for EMGVB with respect to MGVB, the convergence of the LB at a similar level on the test sets and MSE that eventually converge to rather similar values but that in some cases can be quite different at early iterations (which is expected but irrelevant in applications as at $\theta^\star$ the measures are rather analogous. These observations are quantitatively supported by the results in 8, where all the optimizers lead to rather similar estimates and statistics.

A visual inspection of the marginal densities as e.g. in Figures 7 and 5 reveals that in general both EMGVB and MGVB perform quite well compared to MCMC sampling and that the variational Gaussian assumption is quite feasible for all the volatility models. Note that the skew observed e.g. in Figure 7 for the $\omega$ parameters and the non-standard form of e.g. $\psi$ for the FIGARCH models is due to the parameter transformation: VI is applied on the unconstrained parameters $(\psi_\omega, \psi_\phi)$ and such variational Gaussians are back-transformed on the original constrained parameter space where the distributions are generally no longer Gaussian (Figure 5 opposed to Figure 5).

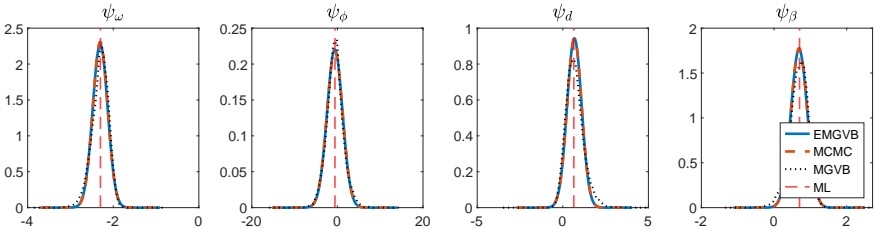

Figure 5: FIGARCH(1,1,1) model. Variational and MCMC marginals for the unconstrained parameters, as a complement to Figure 5 in the main text.

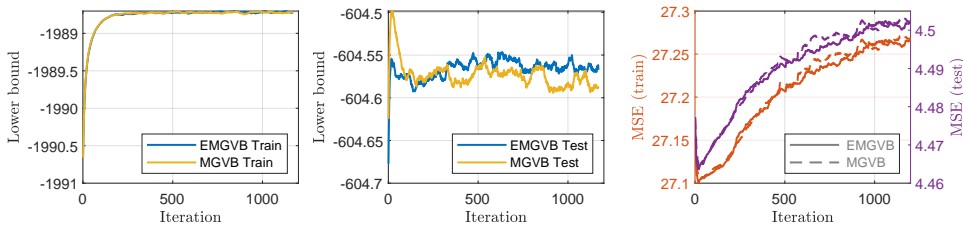

Figure 6: GJR(1,1,1) model. lower bound and mean squared error in the train and test set. Middle and bottom rows.

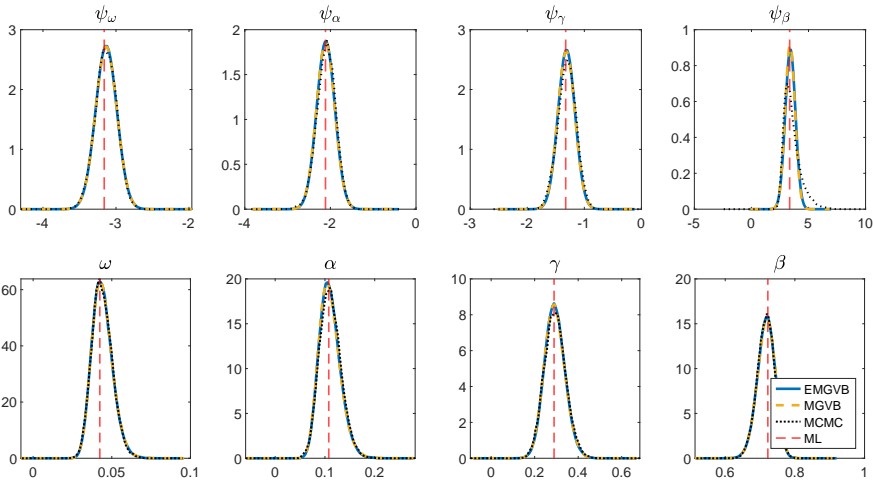

Figure 7: GJR(1,1,1) model. MCMC and variational marginals in for the unconstrained parameters $(\psi_\omega, \psi_\alpha, \psi_\gamma, \psi_\beta)$ (top row) and constrained parameters $(\omega, \alpha, \gamma, \beta)$ (bottom row).

| | $\omega$ | $\alpha$ | $\gamma$ | $\beta_1$ | $\beta_2$ | Train $\mathcal{L}(\theta^\star)$ | $p(y|\theta^\star)$ | MSE | Test $\mathcal{L}(\theta^\star)$ | $p(y|\theta^\star)$ | MSE |
|---|---|---|---|---|---|---|---|---|---|---|---|
| | | | | | | ARCH | | | | | |
| EMGVB | 0.519 | 0.640 | | | | -2247.14 | -2241.03 | 29.818 | -651.11 | -645.75 | 5.447 |
| MGVB | 0.519 | 0.640 | | | | -2247.14 | -2241.03 | 29.818 | -651.11 | -645.75 | 5.447 |
| QBVI | 0.519 | 0.640 | | | | -2247.14 | -2241.03 | 29.815 | -651.11 | -645.74 | 5.445 |
| MCMC | 0.519 | 0.628 | | | | | -2241.01 | 29.694 | | -645.66 | 5.391 |
| ML | 0.519 | 0.640 | | | | | -2241.04 | 29.820 | | -645.73 | 5.447 |
| | | | | | | GARCH(1,0,1) | | | | | |
| EMGVB | 0.043 | 0.230 | | 0.737 | | -2012.40 | -2002.56 | 25.691 | -607.70 | -598.68 | 3.953 |
| MGVB | 0.043 | 0.230 | | 0.737 | | -2012.40 | -2002.56 | 25.693 | -607.69 | -598.67 | 3.954 |
| QBVI | 0.043 | 0.230 | | 0.737 | | -2012.41 | -2002.56 | 25.691 | -607.70 | -598.68 | 3.953 |
| MCMC | 0.042 | 0.226 | | 0.738 | | | -2002.52 | 25.683 | | -598.92 | 3.944 |
| ML | 0.043 | 0.231 | | 0.738 | | | -2002.67 | 25.723 | | -598.43 | 3.957 |
| | | | | | | GJR(1,1,1) | | | | | |
| EMGVB | 0.044 | 0.108 | 0.292 | 0.721 | | -1988.71 | -1976.08 | 27.214 | -604.56 | -592.91 | 4.492 |
| MGVB | 0.044 | 0.108 | 0.293 | 0.721 | | -1988.70 | -1976.09 | 27.243 | -604.56 | -592.90 | 4.497 |
| QBVI | 0.044 | 0.108 | 0.293 | 0.721 | | -1988.70 | -1976.08 | 27.238 | -604.59 | -592.93 | 4.497 |
| MCMC | 0.042 | 0.108 | 0.289 | 0.722 | | | -1976.03 | 27.143 | | -593.29 | 4.476 |
| ML | 0.043 | 0.109 | 0.294 | 0.723 | | | -1976.24 | 27.386 | | -592.58 | 4.515 |
| | | | | | | GJR(1,1,2) | | | | | |
| EMGVB | 0.045 | 0.116 | 0.323 | 0.655 | 0.049 | -1988.39 | -1976.12 | 27.816 | -604.33 | -593.68 | 4.638 |
| MGVB | 0.045 | 0.115 | 0.322 | 0.653 | 0.050 | -1988.38 | -1976.10 | 27.767 | -604.32 | -593.65 | 4.628 |
| QBVI | 0.045 | 0.116 | 0.323 | 0.653 | 0.050 | -1988.39 | -1976.11 | 27.797 | -604.35 | -593.66 | 4.635 |
| MCMC | 0.044 | 0.110 | 0.301 | 0.672 | 0.041 | | -1976.01 | 27.267 | | -593.38 | 4.515 |
| ML | 0.044 | 0.114 | 0.321 | 0.670 | 0.036 | | -1976.12 | 27.778 | | -593.63 | 4.629 |
| | | | | | | EGARCH(1,0,1) | | | | | |
| EMGVB | -0.003 | 0.414 | | 0.929 | | -2033.15 | -2017.67 | 26.840 | -613.73 | -599.43 | 3.931 |
| MGVB | -0.003 | 0.416 | | 0.929 | | -2033.16 | -2017.69 | 26.837 | -613.68 | -599.44 | 3.933 |
| QBVI | -0.003 | 0.416 | | 0.929 | | -2033.16 | -2017.69 | 26.837 | -613.67 | -599.44 | 3.933 |
| MCMC | -0.003 | 0.406 | | 0.932 | | | -2017.64 | 26.858 | | -599.37 | 3.923 |
| ML | -0.003 | 0.415 | | 0.929 | | | -2017.68 | 26.819 | | -599.44 | 3.932 |
| | | | | | | EGARCH(1,1,1) | | | | | |
| EMGVB | -0.015 | 0.350 | -0.171 | 0.930 | | -1995.54 | -1974.97 | 26.193 | -608.06 | -589.08 | 4.128 |
| MGVB | -0.015 | 0.364 | -0.174 | 0.926 | | -1995.69 | -1975.21 | 26.231 | -608.10 | -589.58 | 4.171 |
| QBVI | -0.015 | 0.364 | -0.174 | 0.926 | | -1995.69 | -1975.21 | 26.234 | -608.09 | -589.57 | 4.170 |
| MCMC | -0.015 | 0.340 | -0.169 | 0.932 | | | -1974.92 | 26.165 | | -588.65 | 4.094 |
| ML | -0.015 | 0.350 | -0.172 | 0.929 | | | -1974.98 | 26.208 | | -589.09 | 4.129 |
| | | | | | | EGARCH(1,1,2) | | | | | |
| EMGVB | -0.015 | 0.355 | -0.173 | 0.917 | 0.011 | -1998.67 | -1975.03 | 26.235 | -610.85 | -589.17 | 4.143 |
| MGVB | -0.016 | 0.395 | -0.185 | 0.859 | 0.061 | -1999.55 | -1976.00 | 26.572 | -611.10 | -590.33 | 4.291 |
| QBVI | -0.016 | 0.395 | -0.186 | 0.855 | 0.065 | -1999.55 | -1976.01 | 26.589 | -611.08 | -590.30 | 4.293 |
| MCMC | -0.015 | 0.335 | -0.166 | 0.955 | -0.022 | | -1974.90 | 26.113 | | -588.63 | 4.078 |
| ML | -0.015 | 0.351 | -0.171 | 0.925 | 0.004 | | -1974.99 | 26.214 | | -589.07 | 4.128 |
| | | | | | | FIGARCH(0,1,1) | | | | | |
| | $\omega$ | $\phi$ | $d$ | $\beta_1$ | | | | | | | |
| EMGVB | 0.100 | | 0.648 | 0.418 | | -2007.49 | -2000.06 | 25.673 | -604.27 | -598.35 | 3.978 |
| MGVB | 0.100 | | 0.647 | 0.418 | | -2007.49 | -2000.06 | 25.673 | -604.27 | -598.35 | 3.978 |
| QBVI | 0.100 | | 0.649 | 0.419 | | -2007.50 | -2000.06 | 25.675 | -604.26 | -598.36 | 3.978 |
| MCMC | 0.102 | | 0.657 | 0.432 | | | -2000.02 | 25.722 | | -598.13 | 3.971 |
| ML | 0.100 | | 0.658 | 0.424 | | | -2000.13 | 25.670 | | -598.66 | 3.989 |
| | | | | | | FIGARCH(1,1,1) | | | | | |
| EMGVB | 0.100 | 0.059 | 0.663 | 0.481 | | -2007.62 | -1999.69 | 25.773 | -604.74 | -598.44 | 3.988 |
| MGVB | 0.100 | 0.059 | 0.663 | 0.481 | | -2007.62 | -1999.69 | 25.773 | -604.72 | -598.44 | 3.988 |
| QBVI | 0.100 | 0.059 | 0.663 | 0.480 | | -2007.62 | -1999.69 | 25.771 | -604.73 | -598.44 | 3.988 |
| MCMC | 0.100 | 0.062 | 0.656 | 0.480 | | | -1999.67 | 25.784 | | -598.18 | 3.979 |
| ML | 0.099 | 0.060 | 0.669 | 0.483 | | | -1999.74 | 25.767 | | -598.65 | 3.996 |
| | $\beta_0$ | $\beta_1$ | $\beta_2$ | $\beta_3$ | | HAR | | | | | |
| EMGVB | 1.078 | 0.488 | 0.420 | -0.012 | | -5082.83 | -5060.65 | 24.179 | -1240.01 | -1220.23 | 18.857 |
| MGVB | 1.063 | 0.488 | 0.421 | -0.011 | | -5083.02 | -5060.65 | 24.179 | -1240.30 | -1220.26 | 18.860 |
| QBVI | 1.078 | 0.488 | 0.420 | -0.012 | | -5082.83 | -5060.65 | 24.179 | -1240.00 | -1220.23 | 18.857 |
| EMGVB[†] | 1.085 | 0.491 | 0.421 | -0.016 | | -5085.54 | -5060.68 | 24.179 | -1242.81 | -1220.40 | 18.867 |
| MGVB[†] | 1.067 | 0.493 | 0.420 | -0.016 | | -5085.61 | -5060.70 | 24.180 | -1242.97 | -1220.42 | 18.869 |
| QBVI[†] | 1.084 | 0.490 | 0.421 | -0.015 | | -5085.54 | -5060.67 | 24.179 | -1242.81 | -1220.40 | 18.866 |
| MCMC | 1.067 | 0.489 | 0.420 | -0.011 | | | -5060.64 | 24.179 | | -1220.04 | 18.855 |
| ML | 1.079 | 0.488 | 0.420 | -0.012 | | | -5060.63 | 24.179 | | -1220.03 | 18.857 |

Table 8: Parameter estimates (on the actual constrained parameter space) and statistics on models' performance on the train and test set. [†] denotes the use of the $h$-function gradient estimator.

In this section, we apply EMGVB under different assumptions for the structure of the variational covariance matrix. We use the Istanbul stock exchange dataset of Akbilgic et al. (2014), (details are provided in C.4 and Table 11). To demonstrate the feasibility of the block-diagonal estimation under the mean-field framework outlined in Appendix A.7, we model the Istanbul stock exchange national 100 index (ISE):

$$\text{ISE}_t = \beta_0 + \beta_1\text{SP}_t + \beta_2\text{NIK}_t + \beta_3\text{BOV}_t + \beta_4\text{DAX}_t + \beta_5\text{FTSE}_t + \beta_6\text{EU}_t + \beta_7\text{EM}_t + \varepsilon_t$$

with $\varepsilon_t \sim \mathcal{N}\left(0, \sigma^2\right)$ and the covariates respectively correspond to the S&P 500 index, Japanese Nikkei index, Brazilian Bovespa index, German DAX index, UK FTSE index, MSCI European index, and MSCI emerging market index. We estimate the coefficients $\beta_0, \ldots, \beta_7$ and the transformed parameter $\psi_\sigma = \log(\sigma)$, from which $\sigma$ (standard error of the disturbances) is computed as $\sigma = \exp(\psi_\sigma) + \hat{\text{Var}}(\psi_\sigma)/2$, with $\hat{\text{Var}}(\psi_\sigma)$ read from the variation posterior covariance matrix, while for ML regression corresponds to the residuals' root mean squared error.

We consider the following structures for the variational posterior: (i) full covariance matrix (*Full*), (ii) diagonal covariance matrix (*Diagonal*), (iii) block-diagonal structure with two blocks of sizes $8 \times 8$ and $1 \times 1$ (*Block 1*) and, (iv) block diagonal structure with blocks of sizes $1 \times 1$, $3 \times 3$, $2 \times 2$, $2 \times 2$ and $1 \times 1$ (*Block 2*). Case (iii), models the covariance between the actual regressors but ignores their covariance with the regression standard error. Case (iv) groups in the $3 \times 3$ block the indices traded in non-European stock exchanges, and in the remaining $2 \times 2$ blocks the indices referring to European exchanges and the two MSCI indexes. Furthermore, covariances between the intercept and regression's standard error with all the other variables are set to zero. The purpose of this application is that of providing an example for Algorithm 2 and the discussion in Appendix A.7, rather than providing an effective forecasting model supported by a solid econometric rationale. Yet, structures (iii) and (iv) correspond to a quite intuitive grouping of the variables involved in our regression problem, motivating the choice of the dataset.

Tables 9 and 10 summarize the estimation results. Table 9 shows that the impact of the different structures of the covariance matrix is somewhat marginal in terms of the performance measures, with respect to each other and with respect to the ML estimates. As for the logistic regression example, we observe in the most constrained cases (ii) and (iii) certain the estimates of certain posterior means slightly deviate from the other cases indicating that the algorithm perhaps terminates at a different local maximum. Regarding variational covariances reported in Table 10, there is remarkable accordance between the covariance structures (i), (iii) and ML, while for the diagonal structure (ii) and block-diagonal structure (iv) the covariances are misaligned with the ML and full-diagonal ones, further suggesting that the algorithms convergence at different maxima of the lower bound.

From a theoretical perspective, if $\Sigma$ is the covariance matrix of the joint distribution of the eight variates (case (i)), by the properties of the Gaussian distribution, blocks e.g. in case (iv) and diagonal entries should match the corresponding elements in $\Sigma$. It is however not surprising to observe that the elements in the sub-matrices e.g. in cases (ii) and (iv) deviate from those $\Sigma$. Indeed, the results refer to independent optimizations of alternative models (over the same dataset) that are not granted to converge at the same maximum (and thus distribution). Across the covariance structures (i) to (iv) the optimal variational parameters correspond to different multivariate distributions, that independently maximize the lower bound, and that are not constrained to be related to each other. This is indeed confirmed by the differences in the maximized Lower bound $\mathcal{L}\bar{\theta}^\star$ in Table 9 and in the different levels at which the curves in Figure 8 are observed to converge. Thus the blocks in the covariance matrix under case (iv) do not match the entries in $\Sigma$. In this light, the ML estimates' variances in the third panel of Table 10 can be compared to those of case (i), but are misleading for the other cases, as the covariance matrix of the asymptotic (Gaussian) distribution of the ML estimator is implicitly full.

| | $\beta_0$ | $\beta_1$ | $\beta_2$ | $\beta_3$ | $\beta_4$ | $\beta_5$ | $\beta_6$ | $\beta_7$ | $\sigma$ |
|---|---|---|---|---|---|---|---|---|---|
| Full | 0.001 | 0.098 | 0.079 | −0.272 | −0.168 | −0.354 | 1.165 | 0.943 | 0.014 |
| Diagonal | 0.001 | 0.054 | 0.107 | −0.211 | 0.135 | −0.008 | 0.519 | 0.888 | 0.014 |
| Block 1 | 0.001 | 0.098 | 0.079 | −0.272 | −0.167 | −0.353 | 1.164 | 0.944 | 0.014 |
| Block 2 | 0.001 | 0.066 | 0.127 | −0.205 | 0.244 | 0.225 | 0.188 | 0.844 | 0.014 |
| ML | 0.001 | 0.099 | 0.078 | −0.273 | −0.174 | −0.363 | 1.179 | 0.946 | 0.014 |

| | Train | | | Test | |
|---|---|---|---|---|---|
| | $\mathcal{L}(\theta^\star)$ | $p(y|\theta^\star)$ | MSE | $p(y|\theta^\star)$ | MSE |
| Full | 1186.079 | 1223.699 | 0.194 | 316.849 | 0.041 |
| Diagonal | 1177.810 | 1219.055 | 0.198 | 317.042 | 0.041 |
| Block 1 | 1186.082 | 1223.701 | 0.194 | 316.848 | 0.041 |
| Block 2 | 1173.852 | 1213.509 | 0.204 | 316.696 | 0.041 |
| ML | | 1223.728 | 0.194 | 316.871 | 0.042 |

Table 9: Posterior means, ML estimates and performance measures on the train and test set. $\sigma$ refers to the disturbances' standard deviation. Block 1 corresponds to case (iii) and Block 2 to case (iv).

**Full**

| | | $\beta_1$ | $\beta_2$ | $\beta_3$ | $\beta_4$ | $\beta_5$ | $\beta_6$ | $\beta_7$ | $\beta_8$ | $\sigma$ |
|---|---|---|---|---|---|---|---|---|---|---|
| ML | $\beta_0$ | | 0.001 | 0.003 | -0.001 | -0.002 | -0.001 | 0.006 | -0.009 | 0.000 |
| | $\beta_1$ | 0.001 | | -0.046 | -2.970 | -1.858 | -0.789 | -0.186 | 1.603 | 0.009 |
| | $\beta_2$ | 0.003 | -0.032 | | 1.094 | 0.310 | 0.691 | -0.409 | -4.137 | 0.010 |
| | $\beta_3$ | −0.001 | -2.992 | 1.094 | | 0.897 | 0.596 | -0.949 | -4.693 | -0.017 |
| | $\beta_4$ | −0.002 | -1.906 | 0.297 | 0.961 | | 3.687 | -20.246 | -0.538 | -0.032 |
| | $\beta_5$ | −0.002 | -0.729 | 0.679 | 0.622 | 3.836 | | -28.377 | -1.364 | 0.032 |
| | $\beta_6$ | 0.007 | -0.170 | -0.393 | -1.065 | -20.477 | -28.813 | | -3.426 | -0.004 |
| | $\beta_7$ | −0.009 | 1.605 | -4.148 | -4.700 | -0.563 | -1.386 | -3.324 | | 0.017 |

**Block 1**

| | | $\beta_1$ | $\beta_2$ | $\beta_3$ | $\beta_4$ | $\beta_5$ | $\beta_6$ | $\beta_7$ | $\beta_8$ | $\sigma$ |
|---|---|---|---|---|---|---|---|---|---|---|
| Block 2 | $\beta_0$ | | 0.001 | 0.003 | -0.001 | -0.002 | -0.002 | 0.006 | -0.009 | |
| | $\beta_1$ | | | 0.015 | -2.953 | -1.905 | -0.743 | -0.181 | 1.547 | |
| | $\beta_2$ | | 0.142 | | 1.071 | 0.322 | 0.652 | -0.420 | -4.174 | |
| | $\beta_3$ | | -2.981 | -0.364 | | 0.953 | 0.615 | -1.092 | -4.669 | |
| | $\beta_4$ | | | | | | 3.858 | -20.560 | -0.632 | |
| | $\beta_5$ | | | | | -8.490 | | -28.409 | -1.408 | |
| | $\beta_6$ | | | | | | | | -3.195 | |
| | $\beta_7$ | | | | | | | -4.325 | | |

| | $\beta_1$ | $\beta_2$ | $\beta_3$ | $\beta_4$ | $\beta_5$ | $\beta_6$ | $\beta_7$ | $\beta_8$ | $\sigma$ |
|---|---|---|---|---|---|---|---|---|---|
| Full | 0.000 | 5.610 | 3.174 | 5.324 | 16.804 | 27.424 | 52.868 | 15.323 | 1.170 |
| Diagonal | 0.001 | 1.910 | 1.779 | 1.860 | 2.152 | 2.662 | 2.257 | 3.406 | 1.211 |
| Block 1 | 0.000 | 5.599 | 3.199 | 5.361 | 16.996 | 27.320 | 53.149 | 15.278 | 1.184 |
| Block 2 | 0.001 | 4.434 | 2.009 | 3.869 | 8.517 | 11.167 | 4.957 | 7.798 | 1.170 |
| ML | 0.000 | 5.609 | 3.171 | 5.361 | 16.893 | 27.676 | 53.514 | 15.283 | |

Table 10: Covariance matrices. Top table: covariance matrix of the estimated coefficients for the full variational posterior and covariances of the ML estimate. Second and third table: block covariance matrices where Block 1 corresponds to case (iii) and Block 2 to case (iv). Bottom table: variances of the full and diagonal posteriors along with ML variances. All the entries across the tables are multiplied by $10^4$.

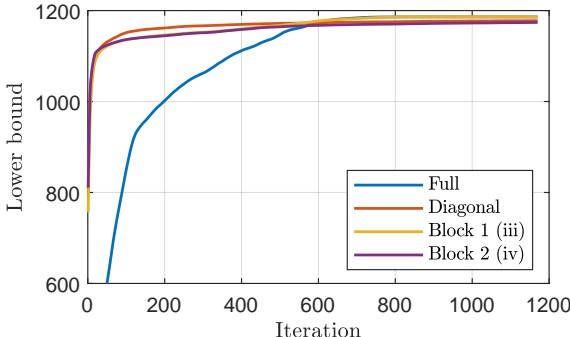

Figure 8: Lower bound optimization for the Istanbul data under different variational covariance structures.

## C.4  DATASETS AND HYPERPARAMETERS

| Dataset | Model | Number of Parameters | Number of samples | Samples in train set | Samples in test set | Period |
|---|---|---|---|---|---|---|
| Labour | Logistic regression | 8 | 753 | 564 (75%) | 189 (25%) | |
| Volatility | ARCH | 2 | 2112 | 1689 (80%) | 423 (20%) | 3-Jan-2014 / 09-Jun-2022 |
| | GARCH(1,0,1) | 3 | 2112 | 1689 (80%) | 423 (20%) | |
| | GJR(1,1,1) | 4 | 2112 | 1689 (80%) | 423 (20%) | |
| | GJR(1,1,2) | 5 | 2112 | 1689 (80%) | 423 (20%) | |
| | EGARCH(1,0,1) | 3 | 2112 | 1689 (80%) | 423 (20%) | |
| | EGARCH(1,1,1) | 4 | 2112 | 1689 (80%) | 423 (20%) | |
| | EGARCH(1,1,2) | 5 | 2112 | 1689 (80%) | 423 (20%) | |
| | FIGARCH(0,1,1) | 3 | 2112 | 1689 (80%) | 423 (20%) | |
| | FIGARCH(1,1,2) | 4 | 2112 | 1689 (80%) | 423 (20%) | |
| | HAR (Linear regr.) | 4+1 | 2102 | 1681 (80%) | 421 (20%) | 4-Feb-2014 / 28-Jun-2022 |
| Istanbul | Linear regression | 8+1 | 536 | 428 (80%) | 108 (20%) | 5-Jan-2009 / 22-Feb-2022 |

| | EMGVB optimizer | | | | | | | | | Initial values | | Prior | |
|---|---|---|---|---|---|---|---|---|---|---|---|---|---|
| Experiment | $\beta$ | Grad clip | Grad clip init. | $\omega$ | $w$ | $t_{\max}$ | $t'$ | $P$ | $S$ | $\mu_1$ | $\Sigma_1$ | $\mu_0$ | $\Sigma_0$ |
| Labour data | 0.01 | 3000 | 1000 | 0.4 | 30 | 1200 | 1000 | 500 | 75 | $\sim \mathcal{N}(0, \Sigma_1)$ | 0.05 | 0 | 5 |
| ARCH-GARCH-GJR | 0.01 | 1000 | 1000 | 0.4 | 30 | 1200 | 1000 | 500 | 150 | ML | 0.05 | 0 | 5 |
| EGARCH | 0.01 | 1000 | 1000 | 0.4 | 30 | 3000 | 2500 | 500 | 150 | ML | 0.05 | 0 | 5 |
| FIGARCH | 0.01 | 1000 | 1000 | 0.4 | 30 | 1200 | 1000 | 500 | 150 | ML | 0.05 | 0 | 5 |
| HAR | 0.001 | 50000 | 500 | 0.4 | 30 | 1200 | 1000 | 500 | 100 | ML | 0.01 | 0 | 5 |
| Istanbul data | 0.07 | 50000 | 500 | 0.4 | 30 | 1200 | 1000 | 500 | 100 | $\sim \mathcal{N}(0, \Sigma_1)$ | 0.01 | 0 | 5 |

Table 11: Details on the datasets and corresponding models, as well as the hyperparameters used in the experiments.

Table 11 summarized some information about the datasets and the setup used across the experiments. For the experiments on the Labour and Volatility datasets, the same set of hyperparameters applies to EMGVB, MGVB (and QBVI). While the Labour[3] and Istanbul[4] datasets are readily available, the volatility dataset is extracted from the Oxford-Man Institute realized volatility library. [5] We use daily close-to-close demeaned returns for the GARCH-family models and 5-minute sub-sampled daily measures of realized volatilities (further annualized) for the HAR model.

---

[3]Publicly available at `key2stats.com/data-set/view/140`. See (Mroz, 1984) for details. The data is also adopted in VI applications e.g. by (Tran et al., 2021b; Magris et al., 2022).

[4]Publicly available at the UCI repository, `archive.ics.uci.edu/ml/datasets/istanbul+stock+exchange`. See (Akbilgic et al., 2014) for details.

[5]Publicly available at `realized.oxford-man.ox.ac.uk`.

# D PROOFS

## D.1 PROOF OF PROPOSITION 1

**Preliminaries:** Noticing that for exponential-family distributions $\mathcal{I}_\zeta = \nabla_\lambda m$ and using the chain rule, $\nabla_\lambda \mathcal{L}(\lambda) = \nabla_\lambda m \nabla_m \mathcal{L}(m) = \mathcal{I}_\zeta \nabla_m \mathcal{L}(\lambda)$, implying that the natural gradient

$$\tilde{\nabla}_\lambda \mathcal{L}(\lambda) = \mathcal{I}_\zeta^{-1} \nabla_\lambda \mathcal{L}(\lambda) = \nabla_m \mathcal{L}(m) \tag{25}$$

can be easily computed as the euclidean gradient with respect to the expectation parameters, without requiring the inverse FIM (Khan & Lin, 2017; Khan et al., 2018a). Khan et al. (2018a) derive

$$\nabla_{m_1} \mathcal{L}(m) = \nabla_\mu \mathcal{L}(m) - 2[\nabla_\Sigma \mathcal{L}(m)]\mu \qquad \text{and} \qquad \nabla_{m_2} \mathcal{L}(m) = \nabla_\Sigma \mathcal{L}(m), \tag{26}$$

which allows expressing the euclidean gradients with the respect to the expectation parameters as euclidean gradients with respect to $\mu$ and $\Sigma$, thus providing an exact relationship between the natural gradients of the LB and its euclidean gradients with respect to the common $(\mu, \Sigma)$ parametrization for $q_\lambda$. Note that the above (and the following) applies to Gaussian distributions only.

**Proof:** The first natural gradient in Proposition 1 is trivial as it follows from the definition of natural gradient and the Gaussian FIM in eq. 1.

If $\xi \equiv \xi(\lambda)$ is a smooth reparametrization of the variational density,

$$\mathcal{I}_\xi = -\mathbb{E}_{q_\xi(\theta)}\left[\nabla_\xi^2 \log q_\xi(\theta)\right] = J\mathcal{I}_\zeta J^\top$$

with $J = \nabla_\xi \lambda$ being the Jacobian matrix (Lehmann & Casella, 1998). If in addition $\xi$ is an invertible function of $\lambda$, then $J$ is itself invertible. Therefore for $\Sigma^{-1} = -2\lambda_2$, the above implies that

$$\mathcal{I}_{\Sigma^{-1}}^{-1} = \left(\nabla_{\Sigma^{-1}}\lambda_2\right)^{-1^\top} \mathcal{I}_{\lambda_2}^{-1}\left(\nabla_{\Sigma^{-1}}\lambda_2\right)^{-1},$$

with $\left(\nabla_{\Sigma^{-1}}\lambda_2\right)^{-1} = \nabla_{\lambda_2}\Sigma^{-1}$. Thus for the natural gradient $\tilde{\nabla}_{\Sigma^{-1}}\mathcal{L}$,

$$\begin{aligned}
\tilde{\nabla}_{\Sigma^{-1}}\mathcal{L} = \mathcal{I}_{\Sigma^{-1}}^{-1}\nabla_{\Sigma^{-1}}\mathcal{L} &= \mathcal{I}_{\Sigma^{-1}}^{-1}\left(\frac{\partial\lambda_2}{\partial\Sigma^{-1}}\frac{\partial\mathcal{L}}{\partial\lambda_2}\right) \\
&= \mathcal{I}_{\Sigma^{-1}}^{-1}\left(\nabla_{\Sigma^{-1}}\lambda_2\right)\nabla_{\lambda_2}\mathcal{L} \\
&= \left(\nabla_{\Sigma^{-1}}\lambda_2\right)^{-1^\top}\mathcal{I}_{\lambda_2}^{-1}\left(\nabla_{\Sigma^{-1}}\lambda_2\right)^{-1}\left(\nabla_{\Sigma^{-1}}\lambda_2\right)\nabla_{\lambda_2}\mathcal{L} \\
&= \left(\nabla_{\lambda_2}\Sigma^{-1}\right)^\top\tilde{\nabla}_{\lambda_2}\mathcal{L} \\
&= -2\tilde{\nabla}_{\lambda_2}\mathcal{L}.
\end{aligned}$$

From eqs. 25 and 26 $\tilde{\nabla}_{\lambda_2}\mathcal{L} = \nabla_{m_2}\mathcal{L} = \nabla_\Sigma\mathcal{L}$, so that

$$\tilde{\nabla}_{\Sigma^{-1}}\mathcal{L} = -2\tilde{\nabla}_{\lambda_2}\mathcal{L} = -2\nabla_\Sigma\mathcal{L},$$

which proves the proposition.

## D.2 GENERAL FORM OF THE EMGVB UPDATE

For a prior $p \sim \mathcal{N}(\mu_0, \Sigma_0)$ and a variational posterior $q \sim \mathcal{N}(\mu, \Sigma)$, by rewriting the LB as

$$\mathbb{E}_{q_\zeta}[h_\zeta(\theta)] = \mathbb{E}_{q_\zeta}[\log p(\theta) - \log q_\zeta(\theta) + \log p(y|\theta)] = \mathbb{E}_{q_\zeta}[\log p(\theta) - \log q_\zeta(\theta)] + \mathbb{E}_{q_\zeta}[\log p(y|\theta)],$$

we decompose $\nabla_\zeta\mathcal{L}$ as $\nabla_\zeta\mathbb{E}_{q_\zeta}[\log p(\theta) - \log q_\zeta(\theta)] + \nabla_\zeta\mathbb{E}_{q_\zeta}[\log p(y|\theta)]$. As in 6, we apply the log-derivative trick on the last term and write $\nabla_\zeta\mathbb{E}_q[\log p(y|\theta)] = \mathbb{E}_{q_\zeta}[\nabla_\zeta[\log q_\zeta(\theta)]\log p(y|\theta)]$. On the other hand, it is easy to show that up to a constant that does not depend on $\mu$ and $\Sigma$

$$\begin{aligned}
\mathbb{E}_{q_\zeta}[\log p(\theta) - \log q_\zeta(\theta)] = &-\frac{1}{2}\log|\Sigma_0| + \frac{1}{2}\log|\Sigma| + \frac{1}{2}d \\
&-\frac{1}{2}\mathrm{tr}\left(\Sigma_0^{-1}\Sigma\right) - \frac{1}{2}(\mu - \mu_0)^\top\Sigma_0^{-1}(\mu - \mu_0),
\end{aligned}$$

so that

$$\nabla_\Sigma \mathbb{E}_{q_\varsigma}[\log p(\theta) - \log q_\varsigma(\theta)] = \frac{1}{2}\Sigma^{-1} - \frac{1}{2}\Sigma_0^{-1},$$

$$\nabla_\mu \mathbb{E}_{q_\varsigma}[\log p(\theta) - \log q_\varsigma(\theta)] = -\Sigma_0^{-1}(\mu - \mu_0).$$

For the natural gradients, we have

$$\tilde{\nabla}_{\Sigma^{-1}} \mathbb{E}_{q_\varsigma}[\log p(\theta) - \log q_\varsigma(\theta)] = -2\nabla_\Sigma \mathbb{E}_{q_\varsigma}[\log p(\theta) - \log q_\varsigma(\theta)] = -\Sigma^{-1} + \Sigma_0^{-1},$$

$$\tilde{\nabla}_\mu \mathbb{E}_{q_\varsigma}[\log p(\theta) - \log q_\varsigma(\theta)] = \Sigma\nabla_\mu \mathbb{E}_{q_\varsigma}[\log p(\theta) - \log q_\varsigma(\theta)] = -\Sigma\Sigma_0^{-1}(\mu - \mu_0),$$

while the feasible naive estimators for $\tilde{\nabla}_\mu \mathbb{E}_{q_\varsigma}[\log p(y|\theta)]$ and $\tilde{\nabla}_{\Sigma^{-1}} \mathbb{E}_{q_\varsigma}[\log p(y|\theta)]$ turn analogous to the right-hand sides of eqs. 9, 10 with $h_\varsigma$ replaced with $\log p(y|\theta)$. This leads to the general form of the EMGVB update, based either on the $h$-function gradient estimator (generally applicable) or the above decomposition (applicable under a Gaussian prior):

$$\tilde{\nabla}_\mu \mathcal{L}(\zeta_t) \approx c_{\mu_t} + \frac{1}{S}\sum_{s=1}^{S}[(\theta_s - \mu_t)\log f(\theta_s)] \tag{27}$$

$$\tilde{\nabla}_{\Sigma^{-1}} \mathcal{L}(\zeta_t) \approx C_{\Sigma_t} + \frac{1}{S}\sum_{s=1}^{S}\left[\left(\Sigma_t^{-1} - \Sigma_t^{-1}(\theta_s - \mu_t)(\theta_s - \mu_t)^\top \Sigma_t^{-1}\right)\log f(\theta_s)\right] \tag{28}$$

where

$$\begin{cases} \begin{cases} C_{\Sigma_t} = -\Sigma_t^{-1} + \Sigma_0^{-1} \\ c_{\mu_t} = -\Sigma_t\Sigma_0^{-1}(\mu_t - \mu_0) & \text{if } p \text{ is Gaussian} \\ \log f(\theta_s) = \log p(y|\theta_s) \end{cases} \\ \\ \begin{cases} C_{\Sigma_t} = 0 \\ c_{\mu_t} = 0 & \text{if } p \text{ is Gaussian or not} \\ \log f(\theta_s) = h_{\zeta_t}(\theta_s) \end{cases} \end{cases} \tag{29}$$

### D.3 Justification for the EMGVB update

For any positive definite matrix $S$ the Riemann gradient $\bar{\nabla}_S\mathcal{L}$, for a differentiable function $\mathcal{L}$, is $S\nabla_S\mathcal{L}S$ (Hosseini & Sra, 2015). This is the form of the Riemann gradient obtained from the SPD (Symmetric and Positive Definite) matrix manifold, for which the following retraction and parallel transport equations for the SPD (matrix) manifold apply:

$$R_S(\xi) = S + \xi + \frac{1}{2}\xi S^{-1}\xi, \text{ where } \xi \in T_S\mathcal{M}, \tag{30}$$

$$\mathcal{T}_{S_t \to S_{t+1}}(\xi) = E\xi E^\top, \quad \text{where } E = \left(S_{t+1}S_t^{-1}\right)^{\frac{1}{2}}, \xi \in T_S\mathcal{M}. \tag{31}$$

with $\xi$ being the rescaled Riemann gradient $\beta\bar{\nabla}_S\mathcal{L} = \beta S\nabla_S\mathcal{L}S$ obtained from the SPD manifold. $\beta > 0$ rescales the tangent vector and is arbitrary. From an algorithmic perspective, $\beta$ is interpreted as a learning rate, driving the magnitude of the gradient component in the retraction. For the precision matrix $\Sigma^{-1}$,

$$\bar{\nabla}_{\Sigma^{-1}}\mathcal{L} = \Sigma^{-1}\nabla_{\Sigma^{-1}}\mathcal{L}\Sigma^{-1}. \tag{32}$$

On the other hand, for the natural gradient

$$\tilde{\nabla}_{\Sigma^{-1}}\mathcal{L} = -2\nabla_\Sigma\mathcal{L} = 2\Sigma^{-1}\nabla_{\Sigma^{-1}}\mathcal{L}\Sigma^{-1},$$

where the first equality comes from Proposition 1 and the second is easy to prove with simple matrix algebra and is furthermore analogous to the form of 2. In this regard, more can be found in (Barfoot, 2020).

The natural gradient is obtained from the Gaussian manifold, so that applying the above SPD manifold retraction and parallel transport equations is technically incorrect. The concept of retraction is general and indeed eq. 4 denotes a generic retraction function $R_\Sigma^{-1}(\cdot)$, whose specific form is

specified in Section 4.2 and coincides with eq. 30. The use of eq. 30, which is specific for the SPD manifold, with the natural gradient obtained from the Gaussian manifold, appears incorrect. The gradient $\tilde{\nabla}_{\Sigma^{-1}}\mathcal{L} = -2\nabla_\Sigma\mathcal{L}$ is not a Riemannian gradient for the SPD manifold but a natural gradient obtained from the Gaussian manifold and equal to $2\bar{\nabla}_\Sigma\mathcal{L}$. The actual exponential map and retraction for updating $\Sigma^{-1}$ based on $\tilde{\nabla}_{\Sigma^{-1}}$ need to be separately worked out for the Gaussian manifold by solving a system of ordinary differential equations, whose curbstone solutions do not coincide with those obtained within the SPD manifold.

In this light, the use of 30 for $\Sigma$ and the natural gradient in Tran et al. (2021a) is not well-justified (Lin et al., 2020). Indeed, their approach can be thought of as inexact in two ways. (i) The natural gradient should read as $2\Sigma\nabla_S\mathcal{L}S$ in place of $\Sigma\nabla_S\mathcal{L}S$ (eq. 5.4 in (Tran et al., 2021a)): the form of retraction and parallel transport therein applied is consistent with the adopted form of the natural gradient ($\Sigma\nabla_S\mathcal{L}S$) which is indeed a Riemann gradient for the SPD manifold, however, the actual natural gradient is $2\Sigma\nabla_S\mathcal{L}S$. (ii) Perhaps the form $\Sigma\nabla_S\mathcal{L}S$ in place of $2\Sigma\nabla_S\mathcal{L}S$ is a typo, thus the application of the SPD-manifold form of the retraction and parallel transport is inexact as it is not applied to a Riemann gradient obtained from the SPD manifold but to the natural gradient obtained from the Gaussian manifold, as also discussed in (Lin et al., 2020).

In this view, our discussion in Section 4 is subject to the same inexact setting as the above case (ii) -a correct form for the natural gradient but mixing manifold structures-, under which the adoption of 30 and 31 is, as in Tran et al. (2021a) not justified (though working in practice).

We now show that the forms of retraction and parallel transport in Section 4.2 are justified and arise from the consistent use of the SPD manifold for updating $2\Sigma^{-1}$, from which the update for $\Sigma^{-1}$ follows, and corresponds to the retraction from in Section 4.2. Later, we show that parallel transport in eq. 15 also applies.

Consider of updating $\left(\mu, 2\Sigma^{-1}\right)$ in place of $\left(\mu, \Sigma^{-1}\right)$. According to eq. 32 the Riemann gradient of $2\Sigma^{-1}$ is for the SPD manifold:

$$\bar{\nabla}_{2\Sigma^{-1}}\mathcal{L} = \left(2\Sigma^{-1}\right)\nabla_{2\Sigma^{-1}}\mathcal{L}\left(2\Sigma^{-1}\right) = \left(2\Sigma^{-1}\right)\frac{1}{2}\frac{\partial\mathcal{L}}{\partial\Sigma^{-1}}\left(2\Sigma^{-1}\right) = 2\Sigma^{-1}\nabla_{\Sigma^{-1}}\mathcal{L}\left(2\Sigma^{-1}\right)$$
$$= \tilde{\nabla}_{\Sigma^{-1}}\mathcal{L}.$$

Thus $2\Sigma^{-1}\nabla_{\Sigma^{-1}}\mathcal{L}\left(2\Sigma^{-1}\right) = \tilde{\nabla}_{\Sigma^{-1}}\mathcal{L}$ is the Riemann gradient w.r.t. $2\Sigma^{-1}$ obtained from the SPD manifold. $2\Sigma^{-1}$ can be updated by the retraction in eq. 30 with the Riemann gradient $\bar{\nabla}_{2\Sigma^{-1}}\mathcal{L}$. The update is now legit and justified, as it consistently adopts the SPD manifold Riemann gradient and the SPD manifold form of retraction:

$$2\Sigma^{-1} \leftarrow R_{2\Sigma^{-1}}\left(\beta\bar{\nabla}_{2\Sigma^{-1}}\right) = R_{2\Sigma^{-1}}\left(\beta\tilde{\nabla}_{\Sigma^{-1}}\mathcal{L}\right),$$

that is,

$$2\Sigma_{t+1}^{-1} = 2\Sigma_t^{-1} + \beta\left[\tilde{\nabla}_{\Sigma^{-1}}\mathcal{L}\right] + \frac{1}{2}\beta^2\left[\tilde{\nabla}_{\Sigma^{-1}}\mathcal{L}\right]\frac{1}{2}\Sigma\left[\tilde{\nabla}_{\Sigma^{-1}}\mathcal{L}\right]. \tag{33}$$

As $\beta > 0$ is arbitrary we can rewrite the above in terms of an arbitrary $\beta' = 2\beta$ (i.e. simple reparametrization of the hyperparameter),

$$2\Sigma_{t+1}^{-1} = 2\Sigma_t^{-1} + \beta'\left[\tilde{\nabla}_{\Sigma^{-1}}\mathcal{L}\right] + \frac{1}{2}\beta'^2\left[\tilde{\nabla}_{\Sigma^{-1}}\mathcal{L}\right]\frac{1}{2}\Sigma\left[\tilde{\nabla}_{\Sigma^{-1}}\mathcal{L}\right]. \tag{34}$$

The update for $\Sigma^{-1}$ follows,

$$\begin{aligned}
\Sigma_{t+1}^{-1} &= \Sigma_t^{-1} + \frac{\beta'}{2}\left[\tilde{\nabla}_{\Sigma^{-1}}\mathcal{L}\right] + \frac{1}{2}\frac{\beta'^2}{4}\left[\tilde{\nabla}_{\Sigma^{-1}}\mathcal{L}\right]\Sigma\left[\tilde{\nabla}_{\Sigma^{-1}}\mathcal{L}\right]\\
&= \Sigma_t^{-1} + \frac{\beta'}{2}\left[\tilde{\nabla}_{\Sigma^{-1}}\mathcal{L}\right] + \frac{1}{2}\left(\frac{\beta'}{2}\right)^2\left[\tilde{\nabla}_{\Sigma^{-1}}\mathcal{L}\right]\Sigma\left[\tilde{\nabla}_{\Sigma^{-1}}\mathcal{L}\right]\\
&= \Sigma_t^{-1} + \beta\left[\tilde{\nabla}_{\Sigma^{-1}}\mathcal{L}\right] + \frac{1}{2}\beta^2\left[\tilde{\nabla}_{\Sigma^{-1}}\mathcal{L}\right]\Sigma\left[\tilde{\nabla}_{\Sigma^{-1}}\mathcal{L}\right] \tag{35}\\
&= R_{\Sigma^{-1}}\left(\beta\tilde{\nabla}_{\Sigma^{-1}}\mathcal{L}\right)\\
&= R_{\Sigma^{-1}}\left(-2\beta\nabla_\Sigma\mathcal{L}\right)
\end{aligned}$$

where $\beta' = \beta/2$. From the above, the main equalities are

$$R_{2\Sigma^{-1}}\big(\beta\nabla^R_{2\Sigma^{-1}}\big) = R_{\Sigma^{-1}}\Big(\beta\tilde{\nabla}_{\Sigma^{-1}}\mathcal{L}\Big) = R_{\Sigma^{-1}}(-2\beta\nabla_\Sigma\mathcal{L}),$$

whose interpretation is as follows. The retraction on $2\Sigma^{-1}$ with the Riemann gradient $2\beta\bar{\nabla}_{2\Sigma^{-1}}$ leads to an update for $\Sigma^{-1}$ (eq. 35) which is the same update obtained with the retraction in 30 on $\Sigma^{-1}$ with $\beta\tilde{\nabla}_{\Sigma^{-1}}\mathcal{L}$. This last gradient is analogous to $-2\beta\nabla_\Sigma\mathcal{L}$ (Proposition 1) and the corresponding retraction for is that presented in Section 4.2.

Blindly updating $\Sigma^{-1}$ with $R_{\Sigma^{-1}}(-2\beta'\nabla_\Sigma\mathcal{L})$ is by itself inexact as it involves the natural gradient from the Gaussian manifold, however, this is equivalent to the update for $\Sigma^{-1}$ that one obtains by in updating $2\Sigma^{-1}$ with the consistent retraction for the SPD manifold $R_{2\Sigma^{-1}}\big(\beta\bar{\nabla}_{2\Sigma^{-1}}\big)$. From the above we also have that, logically, the arbitrary stepsize $\beta$ for updating $\Sigma^{-1}$ is half of that used for updating $2\Sigma^{-1}$, $2\beta$.

A similar argument holds for vector transport. Consider the vector transport for $2\Sigma^{-1}$,

$$T_{2\Sigma_t^{-1}\to 2\Sigma_{t+1}^{-1}}\big(\bar{\nabla}_{2\Sigma^{-1}}\mathcal{L}\big) = E\big(2\beta\bar{\nabla}_{2\Sigma^{-1}}\mathcal{L}\big)E^\top$$

with $E = \Big[\big(2\Sigma_{t+1}^{-1}\big)\big(2\Sigma_t^{-1}\big)^{-1}\Big]^{\frac{1}{2}} = \big[\Sigma_{t+1}^{-1}\Sigma_t\big]^{\frac{1}{2}}$. The vector transport for $\Sigma^{-1}$ is then

$$
\begin{aligned}
T_{\Sigma_t^{-1}\to\Sigma_{t+1}^{-1}}\big(\bar{\nabla}_{\Sigma^{-1}}\mathcal{L}\big) = \frac{1}{2}T_{2\Sigma_t^{-1}\to 2\Sigma_{t+1}^{-1}}\big(\bar{\nabla}_{2\Sigma^{-1}}\mathcal{L}\big) &= E\big(\beta\bar{\nabla}_{2\Sigma^{-1}}\mathcal{L}\big)E^\top \qquad (36) \\
&= E\Big(\beta\tilde{\nabla}_{\Sigma^{-1}}\mathcal{L}\Big)E^\top.
\end{aligned}
$$

Now note that,

$$E\Big(\beta\tilde{\nabla}_{\Sigma^{-1}}\mathcal{L}\Big)E^\top = T_{\Sigma_t^{-1}\to\Sigma_{t+1}^{-1}}\Big(\beta\tilde{\nabla}_{\Sigma^{-1}}\mathcal{L}\Big) = T_{\Sigma_t^{-1}\to\Sigma_{t+1}^{-1}}(-2\beta\nabla_S\mathcal{L}), \qquad (37)$$

as in Section 4.2. The vector transport in the form of eq. 31 is consistently applied to $2\Sigma^{-1}$ based on the Riemann SPD manifold gradient $\bar{\nabla}_{2\Sigma^{-1}}\mathcal{L}$, from which the vector transport for $\Sigma^{-1}$ follows (eq. 36), which equals to the vector transport in eq. 31 applied to $\Sigma^{-1}$ based on the rescaled natural gradient $\beta\nabla_S\mathcal{L} = -2\beta\nabla_S\mathcal{L}$, eq. 37.

### D.4 PROOF OF THEOREM 1

For a Gaussian distribution distribution $q$ with parameter $\zeta = (\mu, \mathrm{vec}(\Sigma))$, be $\nabla_\zeta\mathcal{L}(\zeta_t) = (\nabla_\mu\mathcal{L}(\zeta_t), \mathrm{vec}(\nabla_\Sigma\mathcal{L}(\zeta_t)))$ where $\nabla_x\mathcal{L}(\zeta_t)$ is the derivative of $\mathcal{L}(\zeta)$ with respect to $x$ evaluated at $\zeta = \zeta_t$. Furthermore, adopt the following short-hand notation $q_\zeta := q(\mu, \Sigma)$ and $q_{\zeta_t} := q(\mu_t, \Sigma_t)$.

By using the well-known form of the KL divergence between two multivariate Gaussian distributions, the optimization problem can be written as:

$$
\begin{aligned}
\langle\zeta, \nabla_\zeta\mathcal{L}(\zeta_t)\rangle - \frac{1}{\beta}D_{\mathrm{KL}}(q_\zeta\|q_{\zeta_t}) = {}& \mu^\top\nabla_\mu\mathcal{L}(\zeta_t) + \mathrm{vec}(\Sigma)^\top\mathrm{vec}(\nabla_\Sigma\mathcal{L}(\zeta_t)) \\
& - \frac{1}{2\beta}\left[\log\frac{|\Sigma_t|}{|\Sigma|} - d + \mathrm{tr}\big(\Sigma_t^{-1}\Sigma\big)^\top + (\mu-\mu_t)\Sigma_t^{-1}(\mu-\mu_t)\right] \\
= {}& \mu^\top\nabla_\mu\mathcal{L}(\zeta_t) + \mathrm{tr}(\Sigma\nabla_\Sigma\mathcal{L}(\zeta_t)) \\
& - \frac{1}{2\beta}\left[\log\frac{|\Sigma_t|}{|\Sigma|} - d + \mathrm{tr}\big(\Sigma_t^{-1}\Sigma\big)^\top + (\mu-\mu_t)\Sigma_t^{-1}(\mu-\mu_t)\right].
\end{aligned}
$$

Note that $\nabla_\mu\mathcal{L}(\zeta_t)$ and $\nabla_\Sigma\mathcal{L}(\zeta_t)$ are now constants and that the Hessian of the above equation amounts to the Hessian of the KL divergence. Furthermore the FIM of $q_\zeta$ equals the Hessian of the function $\zeta_t \mapsto D_{\mathrm{KL}}(q_\zeta\|q_{\zeta_t})$ evaluated at $\zeta_t = \zeta$, that is of the above KL divergence:

$$\mathcal{I}_\zeta = \nabla^2_{\zeta_t}D_{\mathrm{KL}}(q_\zeta\|q_{\zeta_t})\big|_{\zeta_t=\zeta}.$$

The multivariate Gaussian distribution is a full-rank exponential family for which the negative Hessian of the log-likelihood (FIM) is the covariance matrix of the sufficient statistics. The FIM is a

convex combination of such positive semi-definite matrices, so it is positive-definite. Thus the above expression is convex with respect to $\zeta_t$.

The objective is optimized by setting its derivatives with respect to $\zeta$ to zero:

$$\nabla_\mu \mathcal{L}(\zeta_t) - \frac{1}{\beta} \Sigma_t^{-1}(\mu - \mu_t) = 0,$$

$$\nabla_\Sigma \mathcal{L}(\zeta_t) - \frac{1}{2\beta} \left[ \Sigma_t^{-1} - \Sigma^{-1} \right] = 0,$$

From which it follows that the optimal updates read

$$\mu = \mu_t + \beta \Sigma_t \nabla_\mu \mathcal{L}(\zeta_t),$$

$$\Sigma^{-1} = \Sigma_t^{-1} - 2\beta [\nabla_\Sigma \mathcal{L}(\zeta_t)]$$

The update for $\mu$ is analogous to the EMGVB update, thus optimal in terms of Theorem 1. On the other hand, we have that the optimal update for $\Sigma^{-1}$ is the above one, which differs from the EMGVB update. Indeed EMGVB accounts for positive-definiteness constraint on $\Sigma^{-1}$, not in the hypotheses of the Theorem. Note however that the EMGVB update for $\Sigma^{-1}$ corresponds to the above one when the second-order term in the retraction (which indeed accounts for the positive definiteness of $\Sigma^{-1}$) is ignored.

