# OpenReview forum: "Exact manifold Gaussian Variational Bayes"
_ICLR.cc/2023/Conference — Submitted to ICLR 2023_

### Official Review · Reviewer_VZUh · 2022-10-20

**Confidence:** 2
**Correctness:** 3
**Technical Novelty And Significance:** 3
**Empirical Novelty And Significance:** 3
**Recommendation:** 6

**Clarity, Quality, Novelty And Reproducibility:**

Clarity: The paper is very well written.
Quality: As far as I can say due to my knowledge of this field, it has quality.
Novelty: As far as I can say, it is novel.
Reproducibility: They do not provide a link to the experiments and code.

**Strength And Weaknesses:**

Strength: Rigurous, state-of-the-art results, style.
Weaknesses: I find the experimental section lack some details in the main manuscript.

**Summary Of The Paper:**

The authors provide a variational inference optimizer on the manifold of positive-definite matrices with all its implementation details and several experiments with results that are, in a wide set of experiments, surprisingly analogous to sampling methods.

**Summary Of The Review:**

I would first like to say that I am not an expert on the variational inference field but of the Bayesian optimization field, so some of my arguments here may be invalid or obsolete. The paper presents a new VI optimizer whose results seem to be the same as some sampling methods, thing that I find very suprising and that if it is validated by VI/MCMC experts it would be very great! The explanation, clarity and quality of the method is also great, so I believe that this can be a great paper.

Some minor things that I can say about the paper to make it better:

In the introduction section, I miss a paragraph that describes the organization of the paper.
In the second section, we find the claim that "Through the problem can be tackled with sampling methods, in high-dimensional apps such as DL, MC techniques are challenging and unfeasible" without references. This may be controversial. In particular, some Bayesian DL methodologies use MCMC sampling and the same happens with Deep Gaussian processes, where I think that this can also be applied successfully BTW (this may be included as a future research line). This needs to be better supported by arguments as it may "bother" some researchers :-)
Provide a reference of the LB maximizers ADAM and company as well as to the beautiful topic of the information geometry of distributions.
The PSD manifold may be better defined as the whole paper depend on this concept, at least the PSD matrix, it may help some novel readers.
IMHO, the related work section must be number 2 to give more cohesion to the paper.
Clarify whether the product operator after equation 6 is a Kronecker product.
"For coherency with the literature available on variational inference and BDL...." provide here a reference.
Provide the complexity of algorithm 1.
In the experiments section: how many repetitions? And the most important thing: provide a link to the experiments.

Thanks for your work and to the other reviewers please validate the method section.

---

### Official Review · Reviewer_8VC5 · 2022-10-22

**Confidence:** 5
**Correctness:** 2
**Technical Novelty And Significance:** 2
**Empirical Novelty And Significance:** 2
**Recommendation:** 3

**Clarity, Quality, Novelty And Reproducibility:**

* Clarity:
Sec 3.1 about the **standard** reinforce gradient estimation should be included in the appendix for a clear presentation.

* Quality:
This submission contains many incorrect statements.
See the weakness section

* Novelty:
I think the only novelty is to introduce the vector transport for the covariance.
However, the vector transport map only valid for a dense/full and diagonal covariance.


**Strength And Weaknesses:**

* Strength
1. Study the method of Tran et al 2021a for black-box variational inference
2. introduce a momentum via the vector transport

* Weakness
 There are many incorrect and misleading statements.
---
1. Ad-hoc usage of the manifold structures. There are two **distinct** types of manifold structures.
Technically speaking, the authors should stick to using one manifold structure at a time.

*  **Matrix Manifold**: Riemannian manifold structure for the positive definite covariance/precision matrix  (the authors use the retraction and the vector transport obtained from the matrix manifold)
*  **Gaussian Manifold**: Riemannian manifold structure for the Gaussian distribution    (the authors use natural/Riemannian gradients  obtained from the Gaussian manifold)

Essentially, the authors compute natural/Riemannian gradients derived from the Gaussian manifold while using the retraction and the vector transport derived from the positive definite matrix manifold.
As discussed by Lin et al 2020., the standard Riemannian metric for the positive definite matrix manifold is **different** from the standard Riemannian metric  (A.K.A., the Fisher information matrix) for the Gaussian manifold.

For any positive definite matrix $P$, the Riemannian gradient (obtained from the matrix manifold) is always $P \nabla_P (\mathcal{L}) P$ as shown in Table 1 of [1].
According to the matrix calculus, we have this identity: $P \nabla_P (\mathcal{L})   P= {\color{red}-  } \nabla_{P^{-1}} (\mathcal{L}) $

* difference 1: For the precision matrix $\Sigma^{-1}$ (when $P=\Sigma^{-1}$),  the Riemannian gradient (obtained from the matrix manifold) is $ {\color{red}-  } \nabla_{\Sigma} (\mathcal{L})$, while the natural gradient (obtained from the Gaussian manifold) w.r.t.  $\Sigma^{-1}$  is $ {\color{red}- 2 } \nabla_{\Sigma} (\mathcal{L})$. See proposition 1 in the main text, where the authors claim this is **exact**/unbiased.

* difference 2: Similarly,  for the covariance matrix $\Sigma$ (when $P=\Sigma$),  the Riemannian gradient (obtained from the  matrix manifold) is $ {\color{red}-  } \nabla_{\Sigma^{-1}} (\mathcal{L})$, while the natural gradient (obtained from the Gaussian manifold) w.r.t.  $\Sigma$  is $ {\color{red}- 2 } \nabla_{\Sigma^{-1}} (\mathcal{L})=2 \Sigma \nabla_{\Sigma} (\mathcal{L})   \Sigma $. See Eq (2) in the main text, where the authors claim this is **inexact**/biased.

* difference 3: The retraction map and the vector transport map for the Gaussian manifold are different from the ones for the matrix manifold.  Why do the authors want to use the two maps (in Sec 3.2) obtained from the matrix manifold, instead? Any justification?

---
2. The natural gradient   (obtained from the Gaussian manifold)  for $\Sigma$ in Eq (2) is indeed exact/unbiased. Thus, the main claim of this submission is incorrect.

There are some caveats when it comes to the Fisher matrix computation as discussed in this blog, https://informationgeometryml.github.io/posts/2021/09/Geomopt01/#caveats-of-the-fisher-matrix-computation

2.1 Technically speaking (as discussed in [2]),  the Fisher information matrix $\mathcal{I}(\Sigma)$ for $\Sigma$ is singular if we consider $\Sigma$ to be a $d$-by-$d$ matrix. Note that by the definition of Gaussian, we have to **explicitly enforce the symmetric constraint** in $\Sigma$ as ${\color{red} \text{highlighted in red}}$ to compute the Fisher information matrix:  $\mathcal{I}(\Sigma) = - E_{p(z)}[ \nabla_\Sigma^2  \log p(z|\mu,\Sigma)] $, where $\log p(z|\mu,\Sigma)=-\frac{1}{2} [d\log (2\pi) + \log \mathrm{det}({ \color{red} \frac{\Sigma+\Sigma^T}{2} } ) + (z-\mu)^T ( { \color{red} \frac{\Sigma+\Sigma^T}{2} }  )^{-1} (z-\mu) ] $.
In other words,  **the Fisher information matrix for $\mathrm{vec}(\Sigma)$ is singular**.  I encourage the authors to work on a 2-dimensional Gaussian case for verification. In this singular case, the **Moore-Penrose inverse** of the Fisher information matrix  (where the authors refer to this inverse as **an inexact approximation**) should be used as discussed in [5].

2.2 A more involved but equivalent way is to consider **the Fisher information matrix for ${\color{red}\mathrm{vech}}(\Sigma)$**, where ${\color{red}\mathrm{vech}}(\Sigma)$ vectorizes symmetric matrix  $\Sigma$ to a $\frac{d(d+1)}{2}$-by-$1$  vector obtained by only the lower triangular part. **The Fisher information matrix w.r.t.  **${\color{red}\mathrm{vech}}(\Sigma)$** is non-singular**.

The authors can also find related discussions in [3].
In summary,  in many existing works ([3-4]) ,  the (**exact**) natural gradient  (obtained from the Gaussian manifold)  w.r.t. $\Sigma$ is exactly Eq 2, which can be computed by either using the the Moore-Penrose inverse for $\mathrm{vec}(\Sigma)$ or using ${\color{red}\mathrm{vech}^{-1}}(\mathbf{a})$,  where $\mathbf{a}$ is a natural gradient w.r.t.  ${\color{red}\mathrm{vech}}(\Sigma)$,

Note that: **the Fisher information matrix for $\mathrm{vec}(\Sigma^{-1})$  is indeed also singular** while the one for $\mathrm{vech}(\Sigma^{-1})$ is not.

---
3. The retraction and vector transport in Sec 3.2 are not exact and only for the full-rank and diagonal matrix manifold instead of the Gaussian manifold.

These two maps are **only valid** for full-rank/dense and diagonal positive definitive matrices.
It is an open problem to find these two maps for (1) the Gaussian manifold and (2) a structured positive definitive matrix manifold (e.g., rank-1 plus diagonal positive definitive matrices).

The authors should note that the retraction and vector transport are often **approximations** of the (exact) manifold exponential map and the (exact) parallel transport map.
 The exponential map and the parallel transport map are defined by solving a system of matrix ODEs, which is hard to obtain a closed-form expression both in the Gaussian case and structured matrix case.

---
References
* [1] Hosseini, Reshad, and Suvrit Sra. "Matrix manifold optimization for Gaussian mixtures." Advances in Neural Information Processing Systems 28 (2015).
* [2] Lin, W., Nielsen, F., Khan, M. E., & Schmidt, M., Introduction to Natural-gradient Descent,  https://informationgeometryml.github.io/posts/2021/09/Geomopt01/#dimensionality-of-a-manifold, https://informationgeometryml.github.io/posts/2021/10/Geomopt02/#definition-of-natural-gradients, and https://informationgeometryml.github.io/posts/2021/12/Geomopt05/#efficient-ngd-for-multivariate-gaussians
* [3] Barfoot, Timothy D. "Multivariate Gaussian variational inference by natural gradient descent." arXiv preprint arXiv:2001.10025 (2020).
* [4] Zhang, Guodong, et al. "Noisy natural gradient as variational inference." International Conference on Machine Learning. PMLR, 2018.
* [5] van Oostrum, Jesse, Johannes Müller, and Nihat Ay. "Invariance properties of the natural gradient in overparametrised systems." Information Geometry (2022): 1-17.

**Summary Of The Paper:**

In this work, the authors consider an "exact" version of Tran et al 2021a in gradient-free settings via the reinforce gradient estimation.
Several standard manifold techniques such as retraction and vector transport are included in the proposed method.
The authors claim that Eq (2) is the natural-gradient w.r.t. $\color{red} \Sigma$ is not exact and propose an exact natural-gradient w.r.t. $\color{red}  \Sigma^{-1}$ in Proposition 1.
Unfortunately, there are many incorrect and misleading statements. In fact, both natural-gradient (Eq (2)) w.r.t. $\color{red} \Sigma$  and natural-gradient (Proposition 1) w.r.t. $\color{red} \Sigma^{-1}$  are exact/unbiased (see pointer 2.1 in the weakness section). Thus, there is no such "exact" version of  Tran et al 2021a.



**Summary Of The Review:**

The main claim of this work: the authors claim that Eq (2) (Tran et al 2021a) is an inexact/biased  natural-gradient w.r.t. $\color{red} \Sigma$, and propose an exact/unbiased natural-gradient w.r.t. $\color{red}  \Sigma^{-1}$ in Proposition 1.

1. Unfortunately, there are many incorrect and misleading statements.  Indeed, Eq (2) (Tran et al 2021a) is the exact/unbiased natural-gradient (obtained from the Gaussian manifold) w.r.t. $\color{red} \Sigma$. The main claim is incorrect.
2. If the authors want to give an "exact" version of Tran et al 2021a, **the authors should give an exact version of natural-gradient w.r.t.  $\color{red} \Sigma$ instead of the one w.r.t. $\color{red}  \Sigma^{-1}$**. Note that natural-gradient decent depends on the choice of parametrization. Although a natural gradient can be non-linearly invariant,  a linear update in the natural gradient (A.K.A., natural-gradient decent) is only linearly invariant (see https://informationgeometryml.github.io/posts/2021/11/Geomopt04//#natural-gradient-descent-is-linearly-invariant )
Thus, the performance of natural-gradient descent depends on the choice of parametrization (e.g., the choice to update either $\color{red} \Sigma$ or $\color{red} \Sigma^{-1}$). In other words, the improvement in the experiments is mainly due to (1) adding the momentum and (2) the change of parametrization.

3. The title, **Exact** manifold **Gaussian** Variational Bayes, is misleading/confusing. It is unclear why the authors use the retraction map and the transport map for the **matrix manifold** instead of the ones for the Gaussian manifold. Moreover, the retraction map is **inexact** even in the full positive definite matrix manifold cases (see Table 1 of [1] for the exact retraction/exponential map).

---

### Official Review · Reviewer_2pY7 · 2022-10-24

**Confidence:** 2
**Correctness:** 3
**Technical Novelty And Significance:** 3
**Empirical Novelty And Significance:** 3
**Recommendation:** 6

**Clarity, Quality, Novelty And Reproducibility:**

Clarity: The submission is easy to follow and written clearly.

Quality: The derivation seems correct, but I did not check the derivation carefully.

Novelty: The method is interesting in my eyes, but I'm not sure about the novelty because I am not an expert in natural gradient.

Reproducibility: I did not check it.


**Strength And Weaknesses:**

Stength: The submission provided a natural gradient method for Gaussian variational inference to guaranttee the positive definite constraint on covariance matrix.

Weakness: I did not find any weakness in the submission.

**Summary Of The Paper:**

The submission proposed a novel natural gradient method for Gaussian variational inference to guaranttee the positive definite constraint on the variational covariance matrix. The propsed method does not need to compute the inverse of FIM explicitly, so it provides exact update rules. Moreover, the proposed method is validated on different models and compared with baseline methods.

**Summary Of The Review:**

The submission proposed a novel natural gradient method for Gaussian variational inference to guaranttee the positive definite constraint on the variational covariance matrix. To avoid the approximation on the computation of the inverse of FIM, the proposed method targets to update the inverse of covariance matrix whose natural gradietn is available in an exact form. Moreover, to facilitate the compuation of gradient of LB, the method also utilizes the log-derivative trick. The method is interesting in my eyes and I am inclined to acceptance, but I would also like to see the opinions of other reviewers as I am not an expert in natural gradient.

---

### Official Review · Reviewer_y1Je · 2022-10-25

**Confidence:** 3
**Correctness:** 4
**Technical Novelty And Significance:** 3
**Empirical Novelty And Significance:** 2
**Recommendation:** 5

**Clarity, Quality, Novelty And Reproducibility:**

My main concern is about the experimental validation. On most metrics, improvements seem to be on the order of a percentage. I'm not convinced that the new method offers a strong improvement against alternatives, though I may have overlooked or misunderstood some parts of the experimentations. In particular, there are no error bars, and the benefits are so tiny, that they could even be statistically insignificant.

Additiionally, I don't see a deep discussion of the computational cost of the new method. Since it is described as being "exact", I would imagine that each step might be computationally more expensive. In that case, does the proposed method strike an interesting tradeoff of computation vs. accuracy, or does it yield small improvements in performance at a significant computational cost. I can't tell from looking at the experiments. I would also say that the scope of the experiments seems limited, but it might be possible to address that by bringing in some material from the appendix.


**Strength And Weaknesses:**

STRENGTHS:
* The paper provides a novel estimator for the gradient in a broad class of models fit with variational inference.
* The derivations and the proposed algorithms are novel, technically sound, and non-trivial. The level of technical depth required to derive the estimator is significant.

WEAKNESSES:
* The experiments do not seem to convey a strong sense of the improvements offered by the new methods. The performance improvements seem marginal at first glance.
* The writing quality and the overall structure of the paper is acceptable, but not quite at the level of polish and organization seen in a typical ICLR accepted paper.

**Summary Of The Paper:**

The authors proposed a new variational inference algorithm for settings where the parameters of the approximate posterior form a Riemannian manifold. Unlike previous methods that relied on various forms approximations, the proposed method is based on an exact formula for the gradient.

**Summary Of The Review:**

I think this paper is currently below the threshold of acceptance, but could be over the threshold after an improved presentation and a more thorough exploration of experimental results.

---

### Decision · Program_Chairs · 2023-01-20

**Decision:**

Reject

**Justification For Why Not Higher Score:**

Authors did not provide a rebuttal, suggesting they agree with the assessment of the reviewers.

**Justification For Why Not Lower Score:**

N/A

**Metareview: Summary, Strengths And Weaknesses:**

One of the reviewers spotted factual errors. These were not disputed by the authors.

**Summary Of Ac-Reviewer Meeting:**

N/A